# Layered Intrusions of Paleoproterozoic Age in the Kola and Karelian Regions

**Valery F. Smol'kin [1], Artem V. Mokrushin [2,\*] and Aleksey V. Chistyakov [3]**

[1] Vernadsky State Geological Museum, Russian Academy of Sciences, Moscow 125009, Russia
[2] Geological Institute—Subdivision of the Federal Research Centre, Kola Science Centre, Russian Academy of Sciences, Apatity 184209, Russia
[3] Institute of Ore Geology, Petrography, Mineralogy, and Geochemistry, Russian Academy of Sciences, Moscow 119017, Russia
\* Correspondence: a.mokrushin@ksc.ru; Tel.: +7-(902)133-39-95

**Abstract:** Large-scale layered intrusions of a peridotite–pyroxenite–gabbronorite complex, to which Cr, Ni, Cu, and PGE deposits and ore occurrences are confined, were emplaced into the Baltic paleocontinent 2.50–2.45 Ga. Layered intrusions in the Monchegorsk Ore District, including the Monchepluton and Imandra–Umbarechka Complex, as well as the gabbro-anorthosite complex of the Main Ridge, were analyzed earlier geochemically and isotopically. In the present paper, the authors analyze layered intrusions in the Kola region (Mount Generalskaya) and Karelia (Kivakka, Kovdozero, and the Burakovsky Pluton). The primary composition of mantle magmas for the layered intrusions is assumed to be identical to that of the komatiitic basalts making up the volcanogenic units of the Vetreny Belt and the Imandra–Varzuga zone. A general model for the formation of layered intrusions includes superplume uplift in the early Paleoproterozoic, the generation of mantle magmas and their injection into the lower portion of the earth crust, the formation of deep-seated and intermediate magma chambers, and the intense contamination of the granulite–metamorphic complex followed by the generation of magma chambers provoked by single or multiple injections.

**Keywords:** Fennoscandian Shield; Kola region; Karelian region; Paleoproterozoic; layered intrusions; mafic–ultramafic; komatiitic basalts; geochemical analysis; magma generation

## 1. Introduction

2.50–2.45 Ga, the uplift of a superplume, the intrusion of abundant mantle magma into the Earth's crust of Baltica Paleocontinent, and the formation of abundant large-scale intrusions of mafic–ultramafic composition with a rhythmically layered internal structure took place [1–5]. The intrusion of abundant high-Mg magma into the lower portion of the earth crust has heated it, resulting in arch formation. This event provoked the formation of rift-related depressions filled with volcanic–sedimentary rocks of Paleoproterozoic age [6].

In the modern erosion section, layered intrusions occur in the Kola–Lapland–Karelian Province, which is the oldest segment of the Fennoscandian Shield (Figure 1). Some of the intrusions host sulfide Cu–Ni–PGE, low-sulfide PGE-metal, chromite, and titanomagnetite ore deposits and occurrences.

Two types of layered intrusions were identified based on geological evidence and the results of U–Pb and Sm–Nd isotope analyses [6–9]. The older (2.53–2.48 Ga) Kola Group consists of several intrusions: Mt. Generalskaya, Ulitoozerskaya, Monchepluton, Pados–Tundra, and the Fedorovo–Pansky Complex. Their intrusion preceded the formation of the Pechenga–Varzuga Belt, which is composed of the sedimentary–volcanic rocks of the Karelian Complex. 2.45–2.43 Ga, the intrusion of the Imandra–Umbarechka Complex, occurring in the Monchegorsk Ore District, and the intrusion of the Lapland–Karelian Group, such as the Olanga Group (Kivakka, Tsipringa, and Lukkulaisvaara), Akanvaara,

Koitelainen, Penikat, Kemi, Portimo, the Koillismaa Complex, and the Burakovsky Pluton, took place under rift-related conditions.

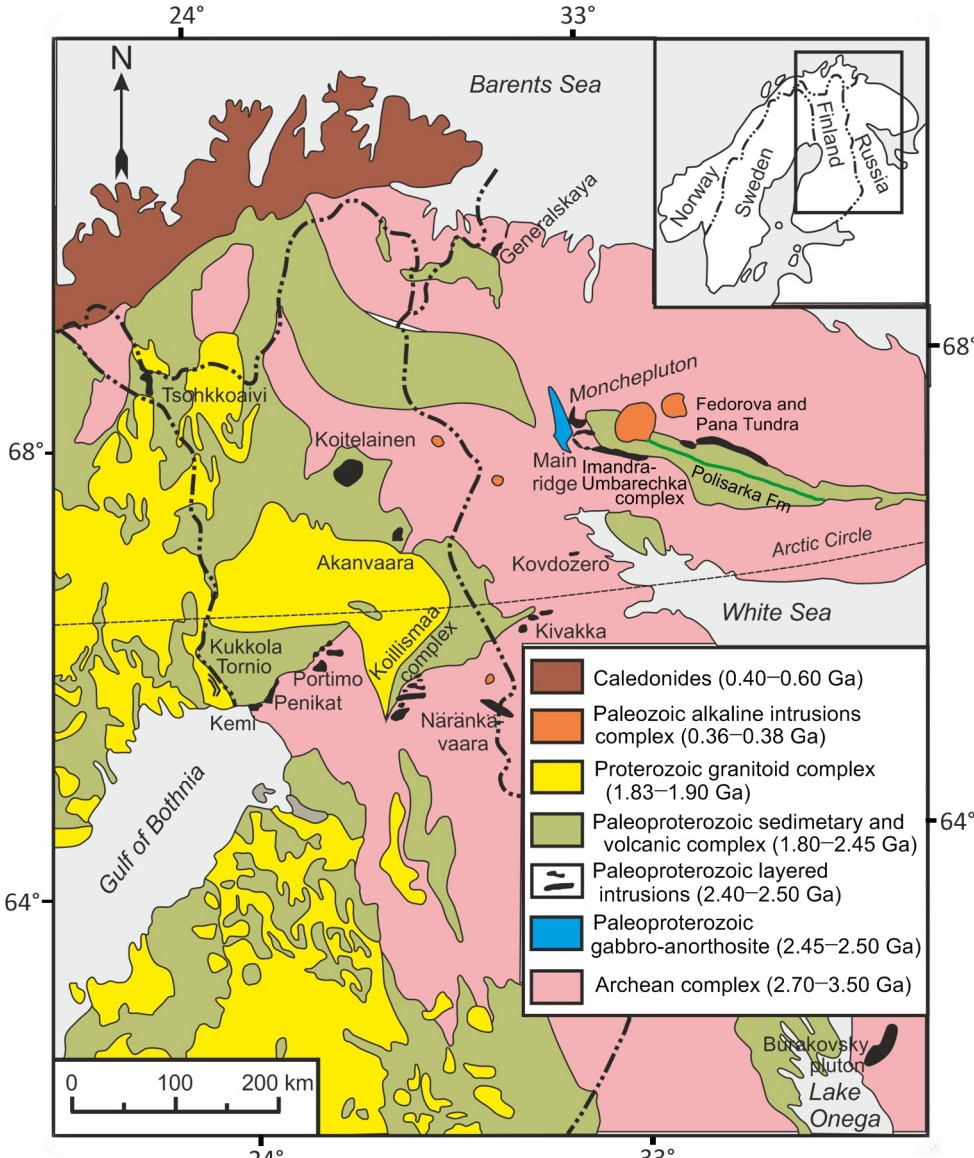

**Figure 1.** Geological map of Fennoscandian Shield (modified after study) [4].

The lherzolite–gabbronorite intrusions of the "Drusite complex", widespread in the western White Sea region, are close in the timing of formation to the latter group [10]. Their formation 2.46–2.43 Ga was provoked by the active migration of enclosing host rock. During the Svecofennian Orogeny, they experienced granulite-facies metamorphism, which gave rise to corona textures. Lying south of the Lapland Granulite Belt are the serpentinized dunite–orthopyroxenite massifs of the Notozerian Complex with chromite mineralization (Pados-Tundra etc.). Their origin has long been the subject of active discussion.

High-Mg magmatism was terminated by the komatiitic basalts of the Polisarka formation in the Imandra–Varzuga zone and the Vetreny Belt formation with a U–Pb age of 2.41 Ga, which are similar in geochemistry to layered intrusion rocks.

The layered intrusions of the Monchegorsk Ore Dictrict, such as Monchepluton, the Imandra–Umbarechka Complex, and the gabbro-anorthosite massifs of the Main Ridge Complex, were previously analyzed [1]. In this paper, the results of whole-rock analysis of layered intrusions, including "Drusite Complex" intrusions, are reported. We have

analyzed reference intrusions of two age groups, such as Mt. Generalskaya, Kivakka, Kovdozero, the Burakovsky Pluton, and Kovdozero, which differ in mineral composition, the degree of mineralization, and differentiation (Figure 1). Available data on komatiitic basalts were used for comparison.

One aim of our study was to reconstruct the formation mechanisms of Paleoproterozoic layered intrusions.

## 2. Geologic Setting

*Mount Generalskaya intrusion* is located in the northwesternmost Kola region, near Luostari Railway Station, at the northern margin of the North Pechenga zone [11]. In the modern erosion section, it covers an area of about 3.5 km × 1.5 km (Figure 2). The intrusion strikes nearly 10–20° N-S and displays a wedge-like shape, an autonomous internal structure, and the eastern and western contacts dipping towards each other at 60–65° and 30–50°, respectively. The upper contact plunges southwest (30–50°) and is overlain by the basal conglomerates of the Televi formation with gabbronorite pebbles and moraine strata. Drilling record shows that intrusion rocks increase in thickness southwestwards from 200–300 m to 1700 m. Several fault systems split up the intrusion into separate blocks.

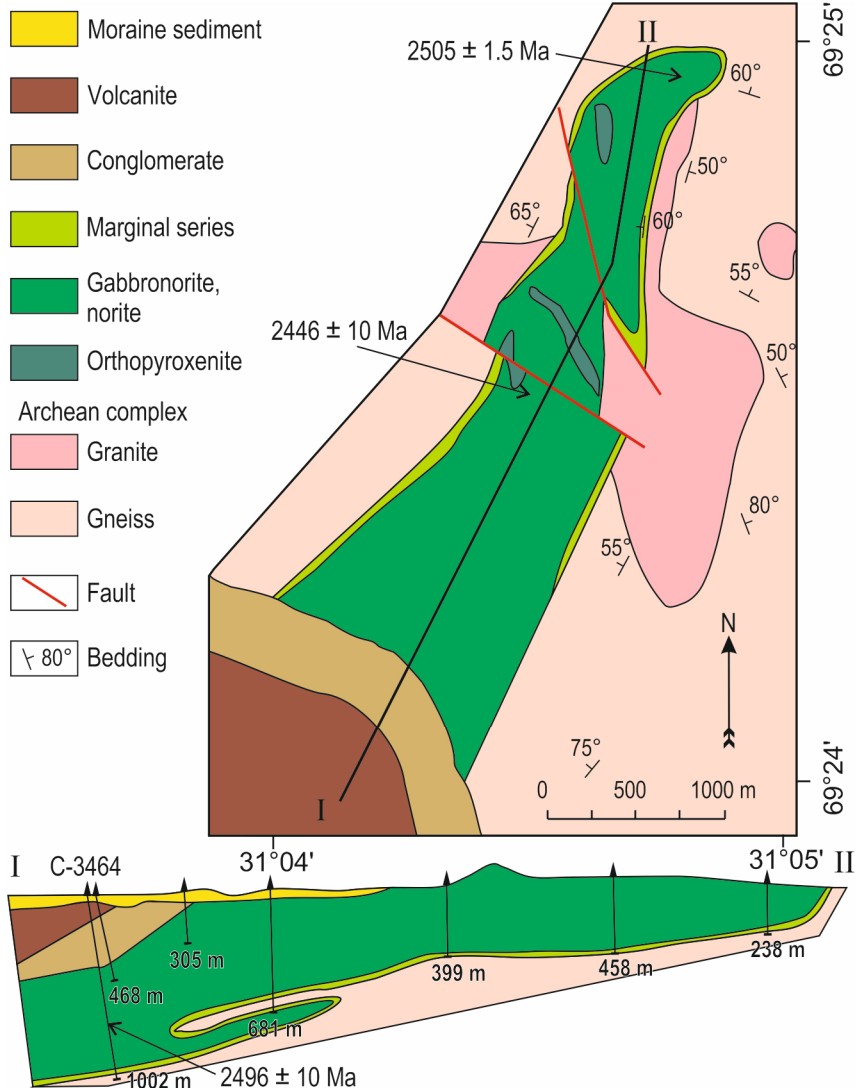

**Figure 2.** Geological map of Mt. Generalskaya intrusion, related geological sections, and the U–Pb age (Ma) of igneous rocks (Table 1) (modified after study) [12].

The intrusion cuts the Neoarchean rocks made up of the tonalite–trondhjemite–granodiorite complex of the Kola–Norwegian Block, triggering their partial melting and the formation of veins of plagiogranite composition at the western contact. Intrusive rocks are cut by quartz metadolerite dykes similar in composition to Mayarvi andesitic basalts in the North Pechenga zone [11].

U–Pb analysis of zircon and baddeleyite showed that the age of gabbronorites from the upper and lower portions of the rock sequence varies from 2505.1 ± 1.6 to 2496 ± 10 Ma (Table 1), suggesting that the intrusion is part of an old group of layered intrusions.

**Table 1.** U–Pb age zircon (ID-TIMS) and Sm–Nd analysis of igneous rocks [13–22].

| Intrusion | Rock | U–Pb (Ma) | Sm–Nd (Ma) | Source |
|---|---|---|---|---|
| Mount Generalskaya | Gabbronorite | 2505.1 ± 1.6 | | [13] |
| | Gabbronorite | 2496 ± 10 | | [14] |
| | Anorthosite | 2447 ± 10 | | [14] |
| | Gabbronorite | | 2453 ± 42 | [15] |
| Kivakka | Gabbronorite | 2445 ± 2 | | [15] |
| | Gabbronorite | 2445 ± 5 | | [16] |
| | Gabbronorite | | 2420 ± 23 | [17] |
| Burakovsky pluton | Gabbronorite | 2449 ± 1.1 | | [17] |
| | Gabbronorite | 2433 ± 4 * | | [18] |
| | Gabbronorite | 2430 ± 4 * | | [18] |
| | Pigeonite-bearing gabbronorite | | 2433 ± 28 | [19] |
| Avdeevskaya dyke | Pigeonite-bearing gabbronorite | | 2436 ± 46 | [19] |
| | Gabbro-pegmatite | 1999 ± 20 ** | | Author |
| Kovdozero | Gabbro-pegmatite | 2436 ± 9 | | [20] |
| Lake Voronii | Anorthosite | 2460 ± 10 | | Author |
| Vetreny Belt, Mt. Golez | Komatiitic basalt | 2405 ± 5 | | [21] |
| Ruiga | Olivine-bearing gabbronorite | 2415 ± 5 | | [22] |

*: Laser ablation multi-collector inductively coupled plasma mass spectrometry (LA-MC-ICP-MS); **: sensitive high-resolution ion microprobe, high-resolution secondary ion mass spectrometry (SIMS-SHRIMP-II).

In addition to magmatic zircon, the rocks contain 2.66–2.61 Ga xenogenic, metamorphic zircon trapped from host rocks. The matrix of overlying conglomerates was found to carry 2.48 Ga detrital zircon consistent with the age of Neuden granites from Northern Norway [23].

Structurally, the intrusion is clearly dominated by gabbronorites, while olivine-bearing gabbronorites, gabbro, anorthosites, norites, orthopyroxenites, and olivine-bearing pyroxenites are less abundant. The rocks are largely metamorphosed under greenschist-facies conditions. Olivine is serpentinized, pyroxenes are partly replaced by tremolite-actinolite-series amphibole, and plagioclase is saussuritized and chloritized. Mildly altered varieties persist only in some blocks, which are mainly located in the central portion of the massif.

Several series in the composite vertical section were tentatively identified [24]. The ca. 100 m thick Lower Marginal Series consists mainly of gabbro-ophite-structured quartz-bearing gabbronorites, as well as less abundant orthopyroxenites, trachytoid micron-sized gabbronorites, and granophyre-bearing gabbronorites. Micron-sized host rock xenoliths which, together with xenogenic zircon, are indicative of contamination, were also found. The intrusion consists mainly of the Layered Series. Its lower (200–250 m) and upper (up to 400 m) portions are dominated by gabbronorites, while its medium portion (350–400 m) is composed of olivine-bearing and olivine-free gabbronorites and norites, gabbro, leucogabbro, and orthopyroxenites. The Layered Series is dominated by mafic rocks with well-defined cumulate structures (pyroxene–plagioclase cumulates), while olivine-

bearing varieties (olivine–plagioclase cumulates), concentrated in the central portion of the layered series, are less abundant. Olivine cumulates, forming single streaks, are scarce. Occurring within the Layered Series are ~4 m thick lenticular anorthosite bodies. Some of the bodies have apophyses extending into underlying gabbronorites. U–Pb analysis has shown that the age of zircon from such anorthosites is 2447 ± 10 Ma (Table 1), suggesting that they are a late vein phase.

Intrusion rocks seem to exhibit variations in the composition of rock-forming minerals, the presence of corona textures occurring as rims at the olivine–plagioclase boundary, an abundance of intercumulus material in olivine-bearing gabbronorite, and an abundance of pigeonite-group pyroxenes (pigeonite and pigeonite–augite) associated with enstatite and augite [24]. The forsterite (*Fo*) content of olivine varies from 68 to 79 mol.%. The ferrosilite (*Fs*) content of orthopyroxene increases from the base upwards from 16 to 26%, while the *Fs* content of clinopyroxene from 9 to 13 mol.%; the anorthite (*An*) content of plagioclase decreases from 78 to 45 mol.%.

The intrusion hosts low-sulfide Cu–Ni–PGE mineralization occurring as variably thick (1 to 20 m) and variably long (0.5 to 1.0 km) horizons [25]. The horizons are concordant with layering and confined to rhythmically layered, taxitic, and poikilitic layered series rocks. Disseminated and veinlet-disseminated types of mineralization occur.

Samples from cores C-3463 and C-3465, intersecting the intrusion over its entire thickness, were analyzed. Analytical data from [24] were also used.

*The Kivakka intrusion* lies in North Karelia, near the eastern boundary of the Paleo-proterozoic Pana–Kuolajärvi structure [11]. It cuts the biotite and amphibole gneisses, migmatites and granite-gneisses making up the Neoarchean complex of the Belomorian Belt. The U–Pb zircon age of the intrusion is 2443 ± 5 Ma; this age is comparable with that of two other intrusions—Tsipringa (2441.3 ± 1.2 Ma) and Lukkulaisvaara (2442.1 ± 1.4 Ma)—located to the north of the Kivakka intrusion. (Table 1; Figure 1).

The Kivakka intrusion occurs as an overturned cone inclined NW at about 40° (Figure 3). Its visible thickness in the central portion is about 2 km. It is a good example of a completely differentiated one-chamber intrusion consisting of olivine and olivine–orthopyroxene to orthopyroxene–plagioclase cumulates. The intrusion is broken into blocks by a system of radial and longitudinal faults.

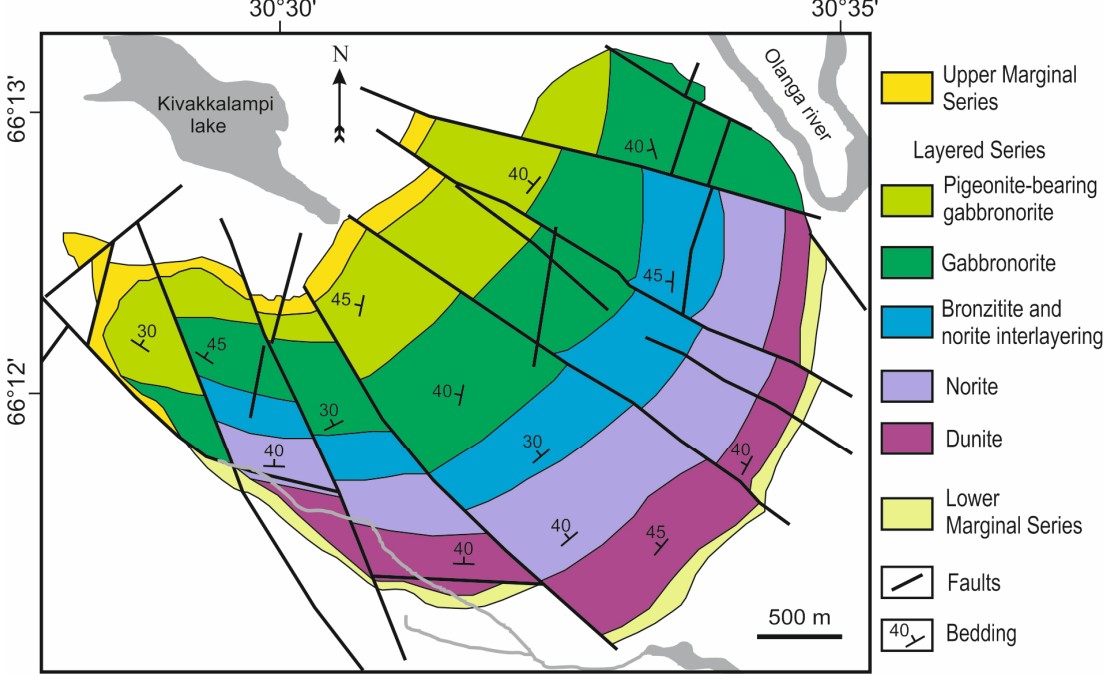

**Figure 3.** Geological map of Kivakka intrusion (modified after study) [26].

According to [26,27], the Lower Marginal, Layered, and Upper Marginal Series (Figure 3) of 100, 1700, and 50 m in average thickness, respectively, are distinguished from the base upwards. The Layered Series, in turn, is divided into a dunite zone (400 m), a harzburgite–melanonorite interlayering zone (400 m), a norite zone (300 m), a gabbronorite zone (320 m), and a gabbronorite zone with pigeonite (320 m).

The Lower Marginal Series consists of medium-grained gabbronorites with a gradual transition through interlayering with olivine-bearing rocks to the Layered Series. Dunites from the dunite zone contain cumulus olivine with 82 to 85 mol.% forsterite (*Fo*). Orthopyroxene and plagioclase, occurring upwards in the dunite zone, become more abundant, triggering a transition to harzburgites and plagioharzburgites. Occurring higher upwards is a harzburgite–melanonorite–orthopyroxenite interlayering zone. The orthopyroxenites are well-defined at the surface and used as a marker horizon.

The norite zone displays rhythmic interlayering of medium-grained meso- and melano-cratic norites. The rhythms are 5–20 m thick. Gabbronorites are less abundant. They contain sulfide platinum mineralization horizons enriched in Pd and Pt bismuth tellurides [28] characteristic of layered intrusions in the Kola–Lapland–Karelian Province.

Occurring higher upwards are medium-grained gabbronorites with equal abundances of bronzite, augite, and plagioclase. They are succeeded by gabbronorites with low-calcium pyroxene–pigeonite, titanomagnetite, and apatite. In the mafic rock unit, the ferrasilite (*Fs*) content of orthopyroxene clearly increases from 16 to 23 mol.%, while the anorthite (*An*) content of plagioclase decreases from 79 to 51 mol.%.

The Upper Marginal Series contains leucocratic rocks and fine-to coarse-grained gabbronorites. The leucocratic rocks occur as lens-shaped bodies made up of coarse pyroxene grains, plagioclase and granophyric quartz, and K-feldspar intergrowths; thus, they can be identified as granophyres. They also contain titanomagnetite, ilmenite, biotite, apatite, zircon, and baddeleyite. Pyroxene and plagioclase are partially replaced by amphibole and saussurite, respectively. According to [27,29], the granophyre bodies were produced by crystallization of fluid-enriched residual melts, suggesting magma differentiation under closed-system conditions.

Samples to be analyzed were taken along the profile extending across the central portion of the intrusion.

The Burakovsky Pluton occurs west of the Onega Depression, East Karelia, in the Vodlozerian Block of the Karelian granite–greenstone domain [30,31]. The pluton covers an area of about 720 km$^2$ and is the biggest layered intrusion in the Kola–Lapland–Karelian Province. It displays a lopolith-like irregular-oval shape and is slightly curved in plan view and elongated in a northeastern direction (Figure 4a). It is 50 km long and 5 to 10 km thick, as indicated by geophysical data [30]. The pluton is hard to study because it is overlain by ~100 m thick moraine strata. Evidence for its internal structure was obtained mainly by drilling 200–300 m deep holes; a few holes were also drilled to depths of 1650 m in the central Aganozero Block (Borehole 20) and 1250 m the southwestern Burakov–Shalozero Block (Borehole 67). A near-N-S-trending fault system split up the pluton into two blocks: Burakov–Shalozero and Aganozero. They showed a cup-shaped internal structure with a gently dipping curvature in the center and steeper margins.

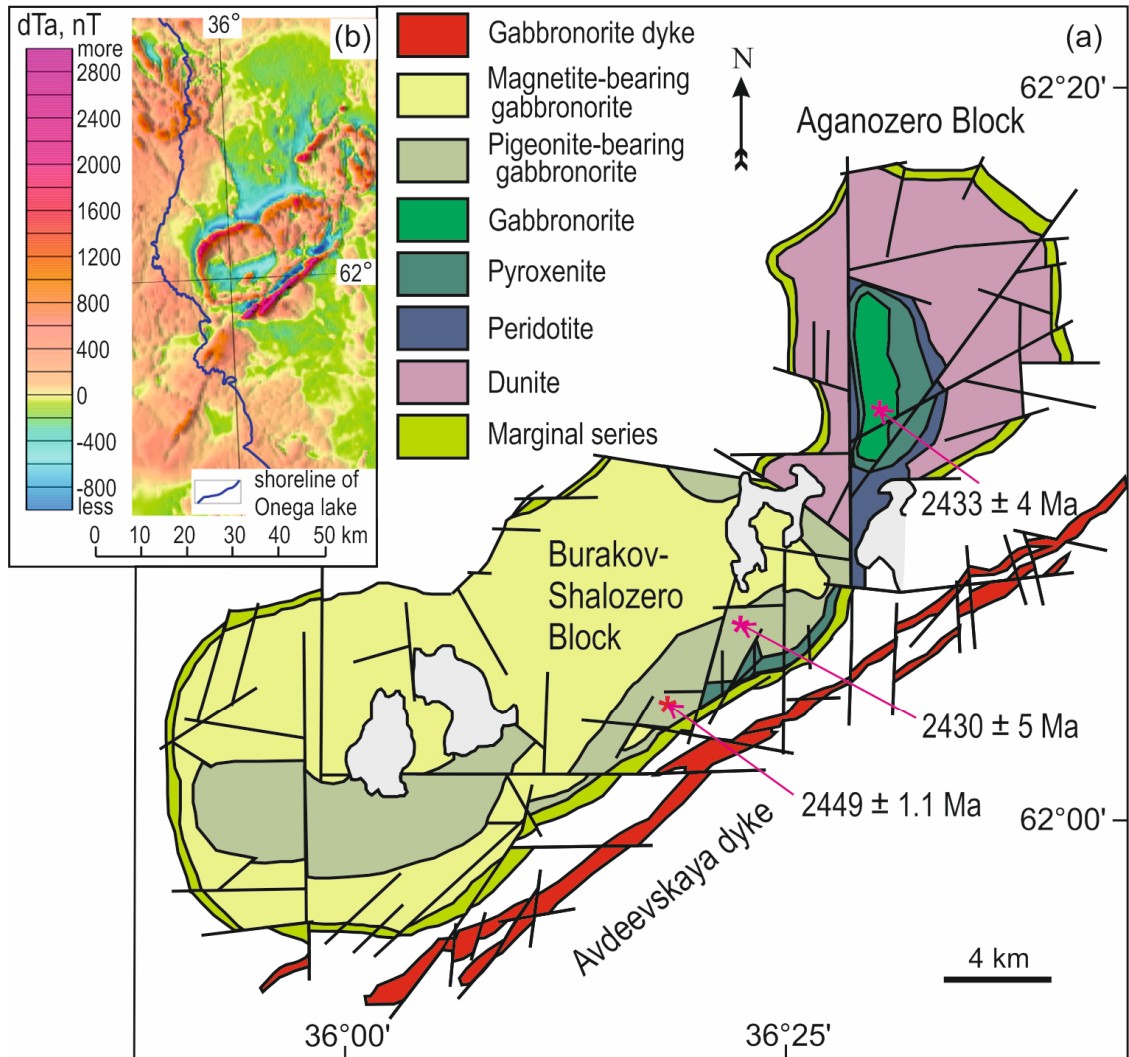

**Figure 4.** Geological map of Burakovsky Pluton (**a**), magnetic field anomalies (**b**), and U–Pb age (Ma) of igneous rocks (Table 1). Geological map modified after study [32].

Two major hypotheses regarding the pluton structure were suggested. According to one hypothesis, both blocks originally existed as a single pluton. The Aganozero Block was then uplifted and eroded [33,34]. According to the other hyuthesis, the blocks originally formed as independent chambers with their upper portions touching each other [32,35,36]. There is no well-defined boundary between the Burakov–Shalozero and Aganozero Blocks on a map of magnetic field anomalies (Figure 4b). It should be noted that both blocks clearly differ in cumulate stratigraphy [32].

Distinguished in the structure of both blocks are Marginal and Layered Series. Marginal Series structures are parallel to the contacts, while the layered intrusions display an autonomous structure relative to the contacts and the dominant sub-horizontal bedding of the layers.

The Marginal Series occurs as a band with gaps along the pluton periphery (Figure 4a,b). It displays a well-defined zonal structure. Occurring directly at the endocontact in the southern portion of the pluton are 0.4–0.9 m thick gabbronorites enriched in micron-sized sulfides, while ~120 m thick amphibolized quartz-bearing norites and gabbronorites occur in the northern portion. They are succeeded by a 10–210 m thick striated zone consisting mainly of fine-to medium-grained leucocratic gabbronorites and separate plagioclase web-sterite layers. Occurring at the internal boundary of the marginal series are serpentinized peridotites with randomly scattered plagioclase websterite layers.

Distinguished in the Layered Series (from the base upwards) are a dunite zone (2500–3000 m), a peridotite zone (thickness 200–400 m), an ore horizon, a pyroxenite zone (20–200 m), a gabbronorite zone (up to 500 m), a pigeonite-bearing gabbronorite zone (up to 850 m), and a magnetite-bearing gabbronorite–diorite zone (up to 1500 m). A complete combination of Layered Series zones is shown for the Burakov–Shalozero Block, while a limited combination is shown for the Aganozero Block (Figure 5).

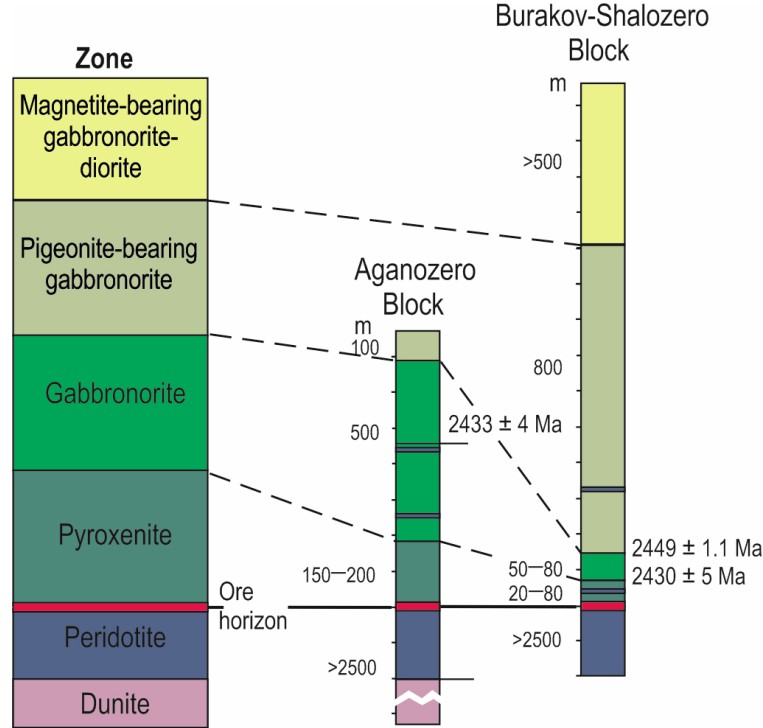

**Figure 5.** Schematic correlation section of Aganozero and Burakov–Shalozero Blocks on Burakovsky Pluton (modified after study) [32] and U–Pb ages (Ma) of ultramafic and mafic rocks (Table 1).

Dunites from the zone of the same name are monomineralic. Peridotites contain ortho- and clinopyroxene, as well as olivine; intercumulus plagioclase and phlogopite are less common. Both zones display rhythmicity and a decrease in olivine concentration from the base upwards. Occurring in the upper portion of the peridotite zone are several ore (chromite) lens-shaped layers that form one Ore Horizon. The thickest layer is a 1.8 to 5.5 m thick Main Ore Horizon. It shows a complex internal structure formed by the alternation of chromitite ores, dunites, peridotites, and pyroxenites. Ores occur as disseminated, densely disseminated and banded types with an average $Cr_2O_3$ concentration of 22.12 wt%.

The overlying (intermediate) pyroxenite zone, showing a maximum thickness of 190–200 m in the Aganozero Block and a minimum thickness of 20–80 m in the Burakov–Shalozero Block, is highly heterogeneous in structure and composition. In the former block, the zone consists mainly of clinopyroxenites and websterites and their olivine-bearing varieties, as well as peridotite (lherzolite, harzburgite) and orthopyroxenite layers. One distinctive feature of the zone is the intensive metasomatic replacement of early magmatic cumulates (olivine, pyroxene) by inverted pigeonite–augite and an abundance of inclusions of quartz–carbonate composition within and between pyroxene grains. In the latter block, the zone is mainly made up of orthopyroxenites, websterites, and olivine-bearing websterites; its middle portion consists of a peridotite marker layer [32].

The gabbronorite zone of the Aganozero Block extends for up to 500 m. Its rhythmic layering is characteristic. The rhythms are 20–185 m thick. Peridotites, websterites, orthopyroxenites, norites, gabbronorites, and leucogabbro are interlayered in the lower portion of the rock sequence; gabbronorites and leucogabbro are interlayered in the upper portion. In

the Burakov–Shalozero Block, the zone consists mainly of gabbronorites; norites, gabbro, and anorthosites occur as layers. The overlying zone is composed of meso- and leuco-cratic gabbronorites with inverted pigeonite and pigeonite–augite. Interstices are filled with quartz, biotite, and K-feldspar aggregate. The pluton sequence is topped by a magnetite-bearing gabbronorite-diorite zone, which occurs only in the Burakov–Shalozero Block. The rocks consist of inverted pigeonite and pigeonite–augite, plagioclase, titanomagnetite, magnetite, and apatite [32].

The forsterite *(Fo)* content of olivine varies from 84 to 90%. The *Fs* content of pyroxenes increases considerably towards the top of the Burakov–Shalozero Block, while the basicity of plagioclase decreases from 52 to 32% *An*.

Lying near the Burakovsky Pluton is the big Avdeevskaya Dyke. It is up to 500 m thick, stretching for 50 km parallel to the southeastern contact of the pluton at a distance of 2–5 km from it (Figure 4). Its southern extension is buried beneath Onega Lake. The dyke is well-defined on a map of magnetic field anomalies (Figure 4b) as several closely spaced dome-shaped bodies dipping sub-vertically southeast. The dyke, exposed over its entire thickness by dimension stone quarries, consists of massive pigeonite-bearing gabbronorites similar in structure, total composition, and mineral composition to Burakov–Shalozero rocks of the same name [37].

Age dates for the Burakovsky Pluton and the Avdeevskaya Dyke, based on U–Pb isotope analysis of zircon and Sm–Nd analysis of rocks and minerals, are scarce. Zircon from the above gabbronorites yielded an age of 2433 ± 4 Ma for the Aganozero Block and 2430 ± 4 and 2449 ± 1.1 Ma for the Burakov–Shalozero Block (Table 1; Figure 5). The results of Sm-Nd analysis of magnetite-bearing gabbronorite–diorite from the Burakov–Shalozero Block (2433 ± 28 Ma, $\varepsilon_{Nd}$ = −3.14) and the Avdeevskaya Dyke (2436 ± 46 Ma, $\varepsilon_{Nd}$ = −1.5) are consistent with these data [19]. Sm–Nd analysis for Aganozero Block leucogabbro dated at 2372 ± 22 Ma suggests additional injection. Local analysis of zircon (SIMS SHRIMP) from gabbro–pegmatite veins cutting the dyke yielded an age of 1999 ± 20 Ma (Table 1). These ages indicate local metamorphism during the Svecofennian Orogeny. The granitic veins cutting intrusive and dyke rocks seem to have been formed in the same period of time.

Widespread at Kovdozero in the Belomorian Belt, SW White Sea region, are massifs of lherzolite–gabbronorite composition known in the geological literature as the "Drusite Complex" [31,38]. Their intrusion was provoked by the active migration of an enclosing rock aged 2.46–2.41 Ga. The massifs occur as small rootless boudin-like bodies resting concordantly with host gneiss schistosity, as well as dykes and relatively large massifs covering an area of up to 80 km². Two types of massifs are distinguished based on mineralogical composition [10]. The former type consists of lherzolite, pyroxenite, olivine-bearing gabbronorite, and norite; the latter type is dominated by norites, gabbronorites, and anorthosites. Magnetite-bearing gabbro–diorites are occasionally encountered. The massif margins are commonly metamorphosed and migmatized; however, rocks with primary magmatic minerals often persist in their central portion.

"Drusite Complex" rocks typically display corona textures in the form of two- or three-layered rims consisting of ortho- and clino-pyroxenes, amphibole, garnet, and symplectic quartz–plagioclase–amphibole aggregates. Recent U–Pb (ID-TIMS) studies have shown that the corona texture and a secondary zircon rim around baddeleyite formed 1.91 Ga were produced by granulite-facies metamorphism during the Svecofennian Orogeny. Assessment of the P–T conditions of formation for bipyroxene's corona textures yields a temperature of 680–900 °C and a pressure of 6.5–8.5 kbar [39].

"Drusite Complex" massifs are similar to layered intrusions [10], as indicated by their similar age, comparable rock composition, similar differentiation trends, and sulfide platinum–metal ore specialization.

The Kovdozero massif, studied earlier in detail by A. Yefimov [40], was chosen for analysis. It is located in the southern Murmansk region, near Zarechensk, on the north shore of Lake Kovdozero. It occurs in the Archean granite–gneiss field of the Belomorian Belt (Figure 1). The ~5 km wide massif is traceable in a near-E–W direction over about

20 km. U–Pb isotope analysis of zircon from gabbro–pegmatite has shown that it formed 2436 ± 9 Ma (Table 1). Anorthosites from Voroniy Island in Kandalaksha Bay formed 2460 ± 10 Ma (Table 1).

The Kovdozero massif is a lenticular body that looks like a trough in a vertical section (Figure 6). It is broken by tectonic dislocations into several blocks composed of various rocks. The biggest blocks are known as Varba, Puakhta, and Yakushikha. Tentatively distinguished in the generalized sequence of the intrusion are the Lower Marginal and the central Layered Series. Fragments of the Lower Marginal Series are present in all the blocks. Their external portion is made up of 1 to 10 m thick plagioclase-bearing orthopyroxenites, while the internal portion consists of a ~150 m thick olivine gabbronorite unit. Occurring in the Lower Marginal Series are irregular gabbro-pegmatite (Puakhta Block) and fine-grained gabbronorite bodies with quartz, granophyre, and biotite (Varba and Yakushikha blocks). The two latter blocks contain orthopyroxenite autobreccia (Yakushikha Block). On an island, lying east of the Puakhta Block, Marginal Series rocks are cut by plagiomicrocline granite veins. Sulfide platinum mineralization of a disseminated type occurs occasionally in the Lower Marginal Zone.

The ~650 m thick Layered Series consists of plagioclase-bearing harzburgites, lherzolites, and orthopyroxenites, olivine-containing and olivine-free norites, and gabbronorites with gradual transitions in between rock type. Lherzolites, olivine norites, gabbronorites, and norites are most common, while troctolites are less abundant. The stratigraphic rock sequence, affected by tectonic dislocations, varies from one block to another. Flank rocks are highly metamorphosed; thus, their primary composition is hard to assess.

The forsterite (*Fo*) content of olivine occurring as a cumulus mineral varies in a harzburgine–lherzolite–olivine-bearing norite and gabbronorite series from 75 to 88 mol.% [40,41]. Orthopyroxene is formed of two morphological types: primary magmatic and reaction. The ferrosilite (*Fs*) content of a primary magmatic type varies from 11 to 18 mol.%. In fine-grained gabbronorites from the Lower Marginal Series, it increases to 20–29 mol.%. The ferrosilite (*Fs*) content of a reaction type of orthopyroxene ranges from 14 to 20 mol.%. Primary magmatic orthopyroxene contains more Al, Ti, and Cr and shows a higher Ca/Ca + Mg + Fe ratio (over 0.28; it is less than 0.1 for a second morphotype 0.1) than a reaction type. These data indicate a difference in pressure upon crystallization [40].

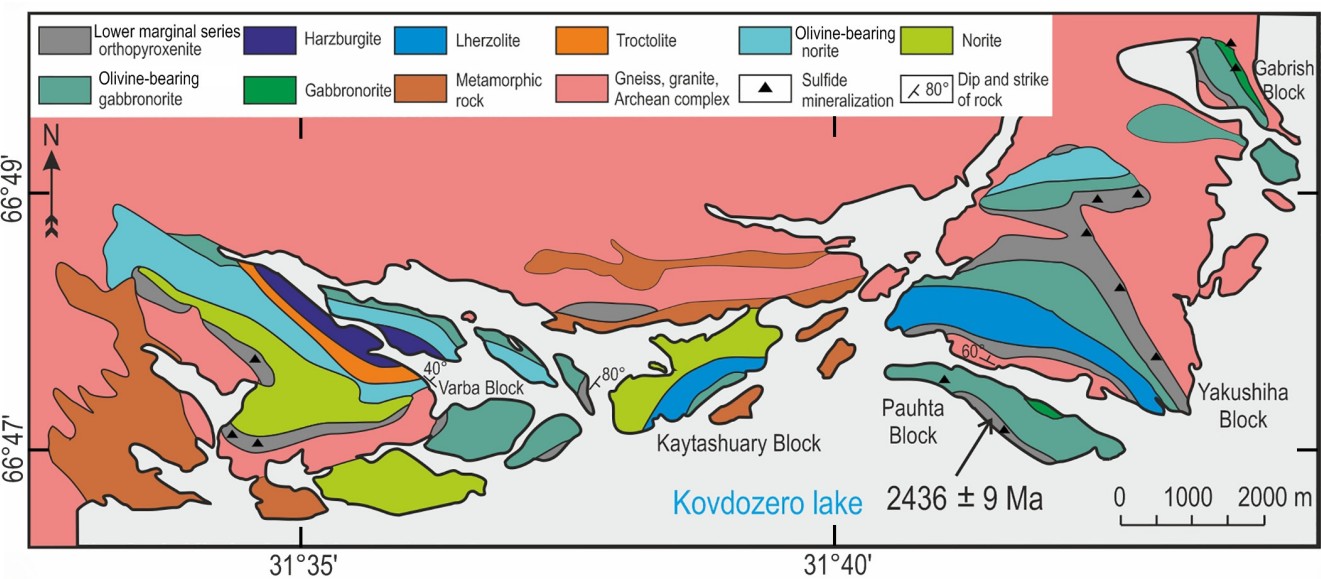

**Figure 6.** Geological map of Kovdozero massif (modified after study) [40] and U–Pb ages (Ma) of ultramafic and mafic rocks (Table 1).

Clinopyroxene follows olivine and orthopyroxene in the crystallization sequence [40]. In Layered Series rocks, it occurs as augite, while in Lower Marginal Series rocks it replaces

orthopyroxene and is consistent in composition with diopside. Plagioclase is an intercumulus phase. The highest anorthite (*An*) content is found in fine-grained rocks in the Lower Marginal Series (61–63 mol.%). This content varies from 26 to 47 mol.% in plagioclase-bearing lherzolites and from 30 to 64 mol.% in mafic rocks. The anomalously low anorthite (*An*) content of brown plagioclase grains could occur due to their post-magmatic alteration.

Unlike most layered intrusions in the Kola–Lapland–Karelian Province, the Kovdozero massif is less differentiated and contains no dunites, chromitites, anorthosites, or near-bottom strata and "Critical horizons" to which sulfide ores are confined.

Rock samples to be analyzed were taken from the Yakushikha Block. Samples from "Drusite complex" massifs on Gorely Island in Kandalaksha Bay and Yudom-Navolok at the Pongoma River mouth in Domashnyaya Bay in the White Sea were also analyzed. They are described in detail in [10].

Komatiitic basalts (low-Ti picrobasalts, high-Mg basalts) make up a large portion of the Polisar formation in the central Kola Peninsula (Figure 1) and the Vetreny Belt formation in East Karelia. The ~800 m thick Polisar formation occurs in the western and central parts of the Imandra–Varzuga zone [42]. It consists dominantly of amphibole– and chlorite–amphibole rocks without relics of primary magmatic minerals. Massive and pillow lavas, as well as tuffs, agglomerate tuffs, and explosive breccias, are distinguished based on the study of relict structural and textural characteristics. Volcanic rock samples were taken from two rock sequences near the Polisarka River mouth and from the upper reaches of the Pana River. Analyses of volcanogenic rocks were published in [1].

The ~4000 m thick Vetreny Belt formation occurs in the northeastern Vetreny Belt. It displays various facies of lava, pillow, diatreme, and hypabyssal types [42–46]. Layered flows with spinifex structures of olivine and pyroxene types, containing relics of olivine, augite, pigeonite, plagioclase (andesine), and alumochromite, are widespread. The degree of differentiation and MgO concentration of komatiitic basalts increase from the southwestern to the northwestern flank (at the Myandukha, Bolshaya Levgora, Olovgora, Shapochka, and Golets mountains). Cumulate olivine with 79–87 mol.% of forsterite (*Fo*) is comparable with olivine from the Burakovsky Pluton (79–88 mol.% *Fo*); however, it is different from olivine in Monchepluton chromitites and dunites (89–95 mol.% *Fo*). Ruiga shallow-depth sub-volcanic intrusion of peridotite–gabbronorite composition is located in the komatiitic basalt field of the Vetreny Belt. Ruiga and other intrusions are spatially close to volcanics [21,22]. The U–Pb zircon age of a layered komatiitic basalt flow on Mt. Golets is 2405 ± 11 Ma, while that of Ruiga Intrusion gabbronorites is 2415 ± 5 Ma (Table 1). The detailed description of major oxides and trace elements is available in the literature [45,46]; the results of isotopic studies are reported in [47].

## 3. Analytical Methods

Rock samples were analyzed mainly at the Central Analytical Laboratory of Karpinsky Russian Geological Research Institute (CIR VSEGEI, St. Petersburg, Russia).

Major and trace elements petrogenic element oxide, as well as Ba, V, and Cr concentrations in rock samples, were assessed using the X-ray fluorescence method (XRF). The method is based on the dependence of the X-ray fluorescent radiation intensity of the chemical element analyzed on its mass fraction in analyzed and calibrated samples in the form of compressed tablets. To excite characteristic fluorescent radiation, an X-ray tube with a rhodium anode mirror, whose characteristic radiation, together with slowing-down radiation, could efficiently excite the atom levels of the elements analyzed, was used. K$\alpha$ lines were used as analytical for all the above elements. The mass fractions of components were calculated using empirical Lucas–Tus coupling equations (multiple regression) describing a relationship between the mass fractions of the component and its fluorescence intensity. A disturbing effect on the element analyzed was taken into account by introducing a corresponding coefficient into the non-linear portion of the equation and measuring the fluorescence intensity of the element. Measurements were made on an ARL-9800 X-ray spectrome-

ter (Switzerland) equipped with fixed spectrometric channels for the above elements and an X-ray tube with a Rh-anode. Quantitative determination boundaries (wt%) were as follows: $Na_2O$ = 0.1–10.0, MgO = 0.2–40.0, $Al_2O_3$ = 1.0–30.0, $SiO_2$ = 2.0–100, $P_2O_5$ = 0.02–5.0, $K_2O$ = 0.1–10.0, CaO = 0.1–50.0, $TiO_2$ = 0.05–5.0, MnO = 0.01–0.5, $Fe_2O_3t$ = 0.2–20.0, Ba = 0.005–0.2, V = 0.001–0.1, and Cr = 0.001–0.5.

Trace and rare earth element (REE) concentrations were calculated using an inductively coupled plasma mass spectrometry (ICP-MS) method on an ELAN-6100 DRCe instrument and TOTALQUANT data processing computer program, including automatic accounting of isotope and molecular superpositions on the mass spectrum analytical lines of the elements analyzed. The samples analyzed were dissolved by pre-alloying with lithium metaborate; the alloy was dissolved in nitric acid for subsequent analysis on the instrument. The sample was processed by alloying with lithium metaborate to fully identify elements, especially REE, present in stable phases. This procedure represents the main difference between alloying and acid decomposition. This advantage was best demonstrated for HREE and other elements present in refractory minerals.

The element compositions of Burakovsky Pluton and Avdeevskaya Dyke rocks were analyzed at the Mineral Substance Analysis Laboratory, IGEM, RAS, Moscow. Major, rare, and trace element oxide concentrations were calculated using an XRF method on a Philips PW 2400 X-ray spectrometer. REE concentrations were measured by ICP-MS on a Plasma Quad PQ2 + Turbo quadrupole mass-spectrometer manufactured by VG Instruments. Detection limits (DL) of elements varied from 1–5 ppb for heavy and medium weight elements (U, Th, REE, and others) to 20–50 ppb for light elements (Be and others). Measurement accuracy accounted for 3–10 rel. % for element contents greater than 20–50 DL.

The results of rock and mineral analyses of the Burakovsky Pluton and Avdeevskaya Dyke were published in [10,32,37].

The results of rock analyses are described in the Supplementary Materials (Tables S1–S3).

## 4. Geochemistry

*Komatiitic basalts.* Variations in major petrogenic components relative to MgO in Polisar formation and Vetreny Belt komatiitic basalts are shown in Figure 7. It follows from their analysis that Vetreny Belt rocks display a much higher degree of differentiation. Polisar formation rocks are poorer in $Al_2O_3$ and CaO, which are feldspar components. The diagram clearly shows two major trends. One trend indicates that $SiO_2$, $Al_2O_3$, CaO, and total ($Na_2O$ + $K_2O$) concentrations increase as MgO concentration decreases. The trend occurs due to olivine phase accumulation, which proceeds when other phases do not accumulate. The other trend indicates no relationship between $Fe_2O_3t$ and MgO, which is possible upon rapid magma crystallization without iron oxidation and ore phase accumulation. Similar trends have been revealed for the sub-volcanic Ruiga intrusion [22]. Considerable variation (0.3 to 1.9 wt%) has been shown for $TiO_2$. Elevated $TiO_2$ concentrations prevail in low-MgO rocks, i.e., late differentiates.

The differentiation degree of the Mt. Generalskaya intrusion is comparable with that of komatiitic basalts (Figures 7 and 8). However, unlike volcanics the intrusion clearly exhibits a direct correlation between $Fe_2O_3t$ and MgO, which is typical of most layered intrusions in the region. Mt. Generalskaya intrusion rocks also contain 2.5 times less $TiO_2$.

*Kivakka* is a one-chamber intrusion with a complete set of rocks from dunites to leuco-gabbro deposited by closed-system differentiation of a single pulse of magma. The rocks have retained their primary outlook because they have not been metamorphosed. Hence, the intrusion could be proposed as a model for the region's layered intrusions. Layered and Marginal Series rocks display varying differentiation trends (Figure 9). The Layered Series exhibits a direct correlation between MgO and $Fe_2O_3t$ and inverse correlations between MgO and $Al_2O_3$ and CaO and ($Na_2O$ + $K_2O$). These characteristics are due to the succession of their major mineral phases with a decline in their crystallization temperature: Ol–(Ol + Opx)–Opx–(Opx + Pl)–(Pl + Opx + Cpx) and the enrichment of rocks in feldspar components ($Al_2O_3$ and CaO, as well as $Na_2O$ and $K_2O$). A sharp bend in the differen-

tiation trend on an MgO – SiO$_2$ curve, indicating mono-mineral rock (orthopyroxenite) crystallization, is also typical of completely differentiated Monchepluton [1]. The Marginal Series clearly shows an inverse differentiation pattern for Al$_2$O$_3$ and CaO, as well as Fe$_2$O$_3$t, and TiO$_2$ accumulation, as indicated by titanomagnetite crystallization. Maximum SiO$_2$, CaO, and (Na$_2$O + K$_2$O) concentrations in Kivakka intrusion rocks and komatiitic basalts are similar (Figures 7 and 9). Al$_2$O$_3$ concentration in intrusive rocks is 5 wt% higher, which seems to be due to higher pressure in the intrusive chamber [27].

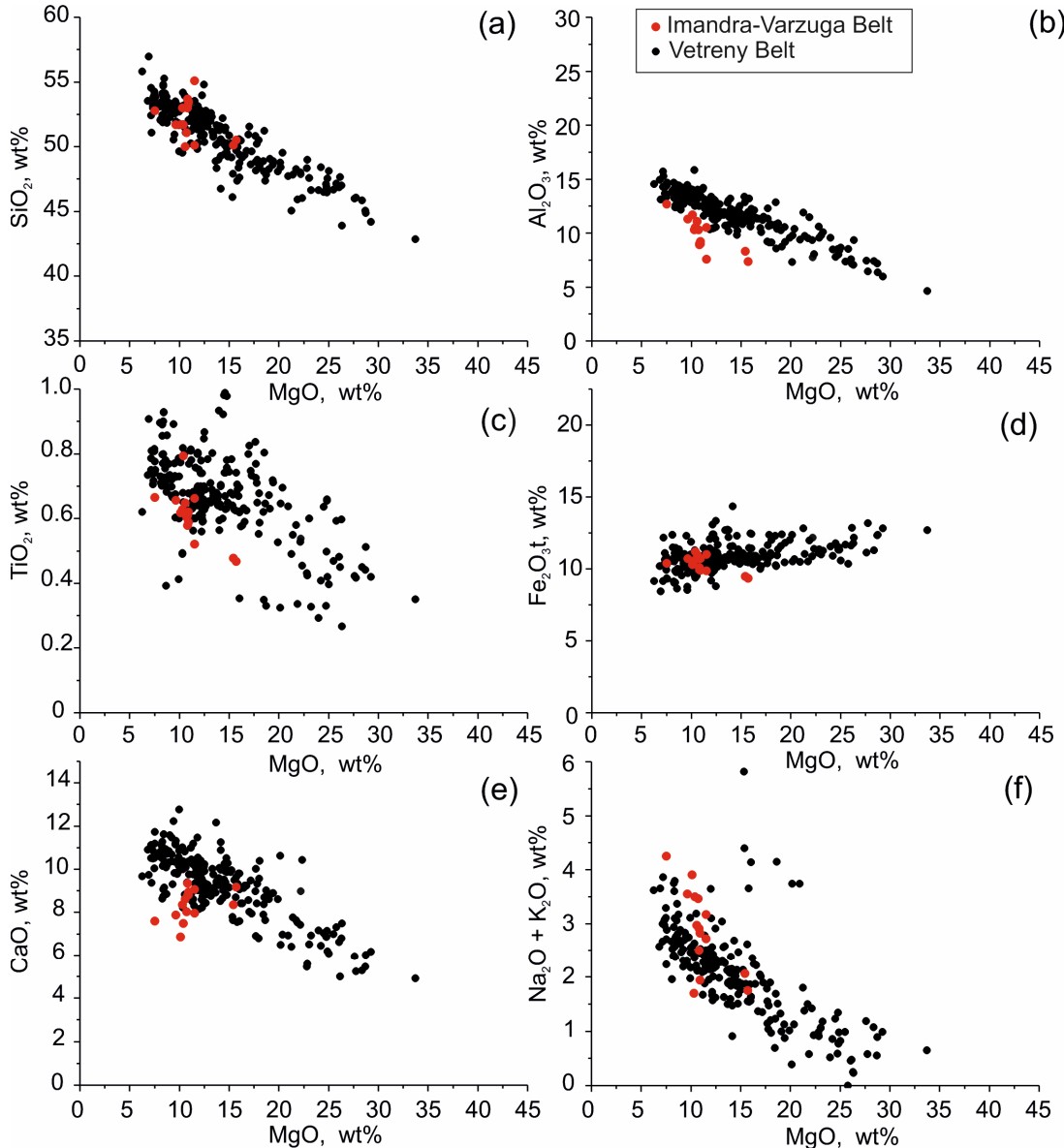

**Figure 7.** Petrochemical diagrams of MgO–SiO$_2$ (**a**), MgO–Al$_2$O$_3$ (**b**), MgO–TiO$_2$ (**c**), MgO–Fe$_2$O$_3$t (**d**), MgO–CaO (**e**), and MgO–(Na$_2$O + K$_2$O) (**f**) systems of komatiitic basalts from the Imandra–Varzuga and Vetreny Belts.

The Burakovsky Pluton is more complex in structure and composition than the Kivakka intrusion, as indicated by the diagrams (Figure 10). In spite of the considerable scattering of composition points, the Burakovsky Pluton retains several major differentiation trends: a direct correlation between MgO and Fe$_2$O$_3$t, as well as inverse correlations between MgO and Al$_2$O$_3$ and CaO, and (Na$_2$O + K$_2$O). The MgO–SiO$_2$ curve shows the sequential crystallization from olivine to orthopyroxene cumulates.

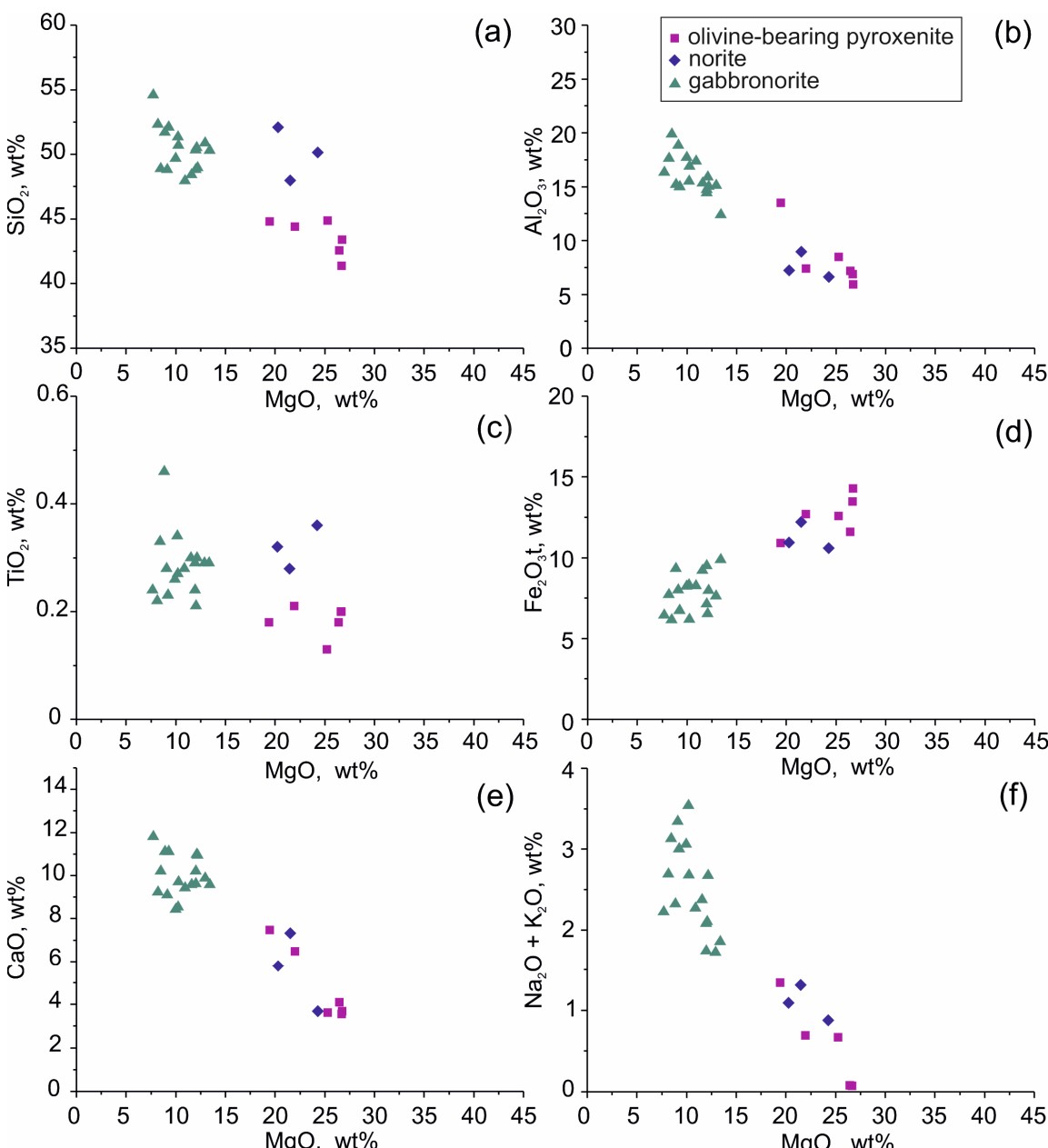

**Figure 8.** Petrochemical diagrams of MgO–SiO₂ (**a**), MgO–Al₂O₃ (**b**), MgO–TiO₂ (**c**), MgO–Fe₂O₃t (**d**), MgO–CaO (**e**), and MgO–(Na₂O + K₂O) (**f**) systems of igneous rocks of Mt. Generalskaya. Authors' and published data were used [24].

Unlike the Kivakka intrusion, the Marginal Series displays a less distinct inverse differentiation trend. The highest Al₂O₃ and CaO concentrations for gabbroic rocks and the heterogeneous distribution of concentrations for TiO₂ and Fe₂O₃t were also revealed. One of positive TiO₂ peaks is related to Burakov–Shalozero Block magnetite-bearing gabbro-diorites, while another peak is related to pyroxenites in the Aganozero Block zone of the same name. The Aganozero and Burakov–Shalozero Blocks differ in the degree and pattern of differentiation processes. Thus, the former contains rocks with MgO concentration in excess of 33 wt% (dunites) and CaO concentration over 13 wt% (clinopyroxenites), which are not present in the latter. Avdeevskaya Dyke rocks are most similar in composition to Burakov–Shalozero gabbronorites with pigeonite.

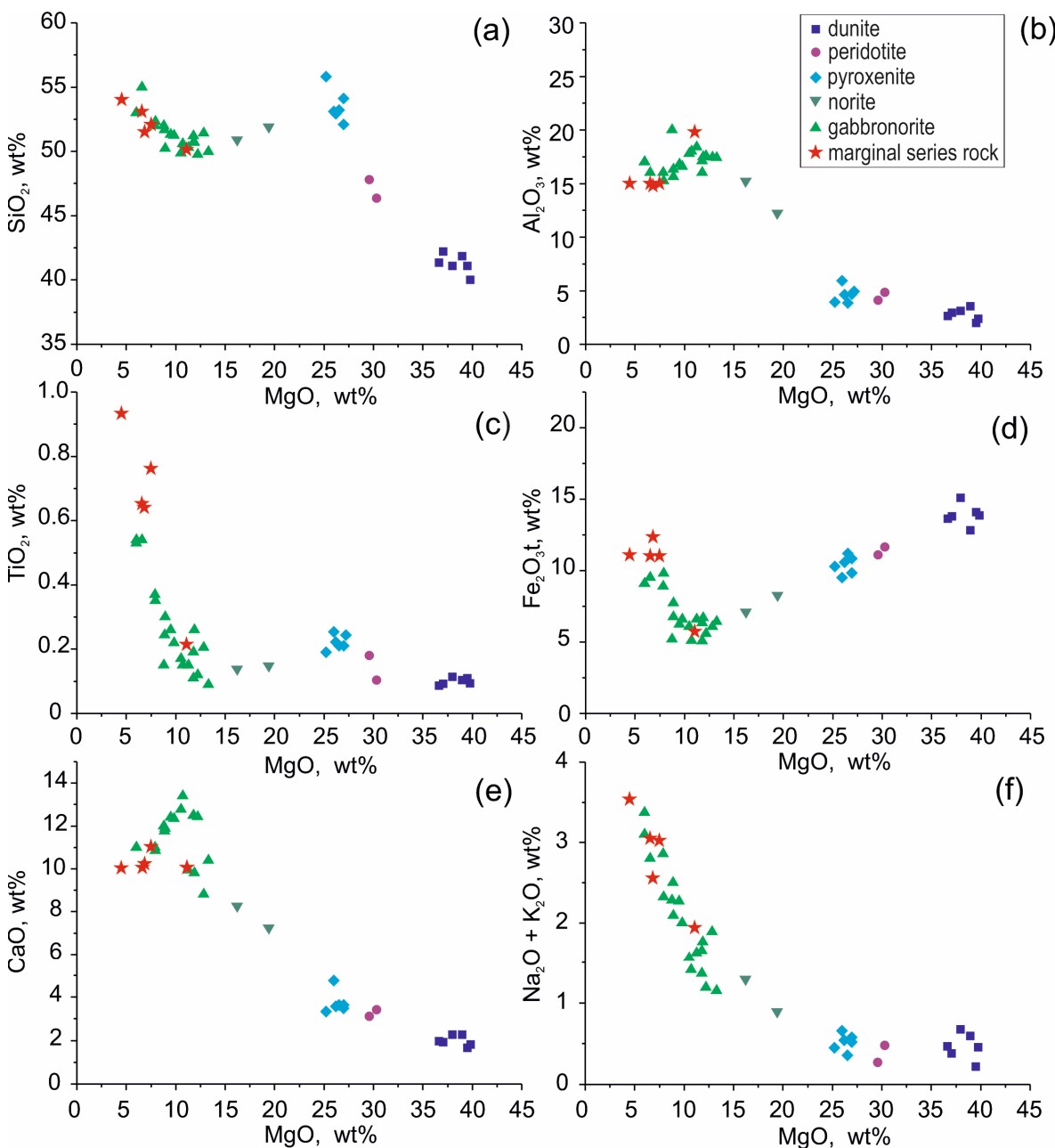

**Figure 9.** Petrochemical diagrams of MgO–SiO$_2$ (**a**), MgO–Al$_2$O$_3$ (**b**), MgO–TiO$_2$ (**c**), MgO–Fe$_2$O$_3$t (**d**), MgO–CaO (**e**), and MgO–(Na$_2$O + K$_2$O) (**f**) systems of igneous rocks of Kivakka. Authors' and published data were used [27].

Variations in the compositions of olivine and orthopyroxene (Figure 11) obey a general pattern: their MgO concentration decreases from dunites to pyroxenites and gabbronorites, with the compositions of olivine from dunites and peridotites from the zones of both blocks overlapping; however, the Burakov–Shalozero Block contains Mg-richer olivine. The greatest differences were revealed between olivine in ore horizons because Aganozero Block olivine contains more MgO with up to 90 mol.% forsterite (*Fo*). Olivine of similar composition was previously reported from Monchepluton chromitite ores [8]. Differences exist also between the compositions of olivine from the peridotites of marker horizons because olivine from the Burakov–Shalozero Block contains more MgO than that from the Aganozero Block. Orthopyroxene from Burakov–Shalozero Block rocks displays greater variations in composition, which is especially significant for gabbronorites.

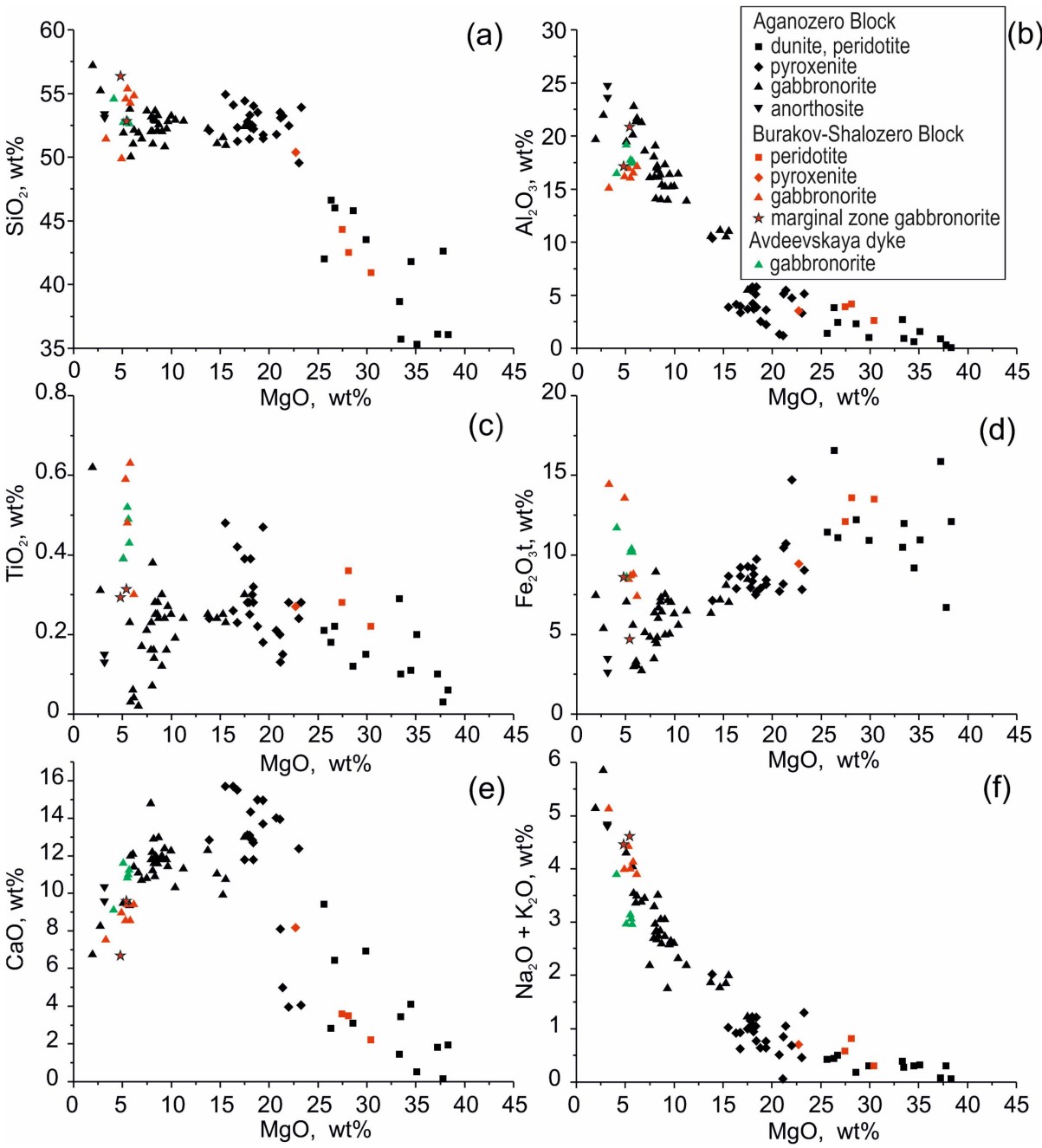

**Figure 10.** Petrochemical diagrams of MgO–SiO$_2$ (**a**), MgO–Al$_2$O$_3$ (**b**), MgO–TiO$_2$ (**c**), MgO–Fe$_2$O$_3$t (**d**), MgO–CaO (**e**), and MgO–(Na$_2$O + K$_2$O) (**f**) systems of igneous rocks of Burakovsky Pluton.

The diagram (Figure 12) shows the composition of rocks from the Kovdozero and other massifs (Gorely and Yudom–Navolok islands) occurring as variably metamorphosed tectonic fragments. The "Drusite Complex" rocks analyzed typically display a well-defined negative trend between MgO and SiO$_2$ plus Al$_2$O$_3$, CaO, and (Na$_2$O + K$_2$O); however, there is no bend on the MgO–SiO$_2$ curve and no correlation between MgO and Fe$_2$O$_3$t. A combination of these characteristics is more typical of komatiitic basalts (Figure 7). The "Drusite Complex" massifs studied seem to be sub-volcanic bodies in which crystallization and differentiation proceeded like those of shallow-depth intrusions. The persistence of a major differentiation pattern in "Drusite Complex" rocks is an argument in favor of the isochemical pattern of multiple amphibolite- and granulite-facies metamorphism.

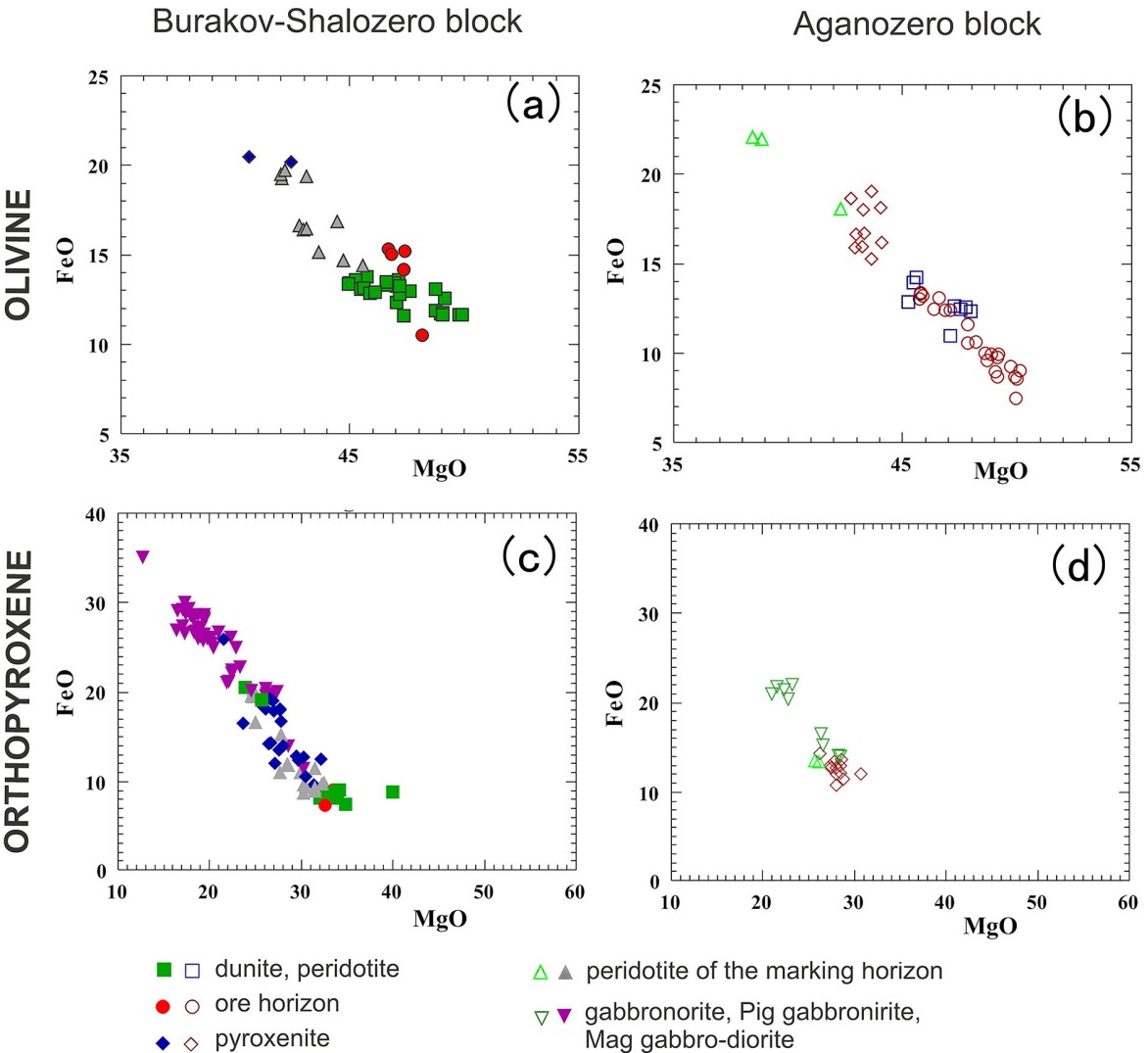

**Figure 11.** Variations of olivine (**a**,**b**) and orthopyroxene (**c**,**d**) composition from the Burakov-Shalozero and Aganozero Blocks.

Some transitional metals contributing to ore formation, e.g., Ni, exhibit a well-defined direct correlation in MgO and between each other for both volcanic and intrusive rocks (Figures 13 and 14). These data indicate that the crystallization of the main silicate phase is a major factor of its distribution because Ni is part of olivine. The absence of correlation between Ni/Co and Fe# also indicates that an ore sulfide phase does not affect silicate magma crystallization (Figure 14b).

The trace elements of komatiitic basalts display the same type of poorly differentiated chondrite-normalized REE pattern with a slight excess of LREE over HREE ($La_n/Lu_n = 1.5$), as well as a poorly defined negative Eu anomaly (Figure 15a).

To focus on the REE abundance of intrusive rocks, the results of REE analysis in the main mineral phases of the Kivakka intrusion are used [27]. Maximum normalized concentrations were shown for augite with equal amounts of LREE and HREE and a positive Sm anomaly, as well as medium concentrations for orthopyroxene with an excess of HREE over LREE ($La_n/Lu_n = 22$) and poorly defined Ce and Eu anomalies. The contrasting pattern of the REE spectrum is shown for mafic plagioclase with a considerable excess of LREE over HREE ($La_n/Lu_n = 300$) and a well-defined positive Eu anomaly.

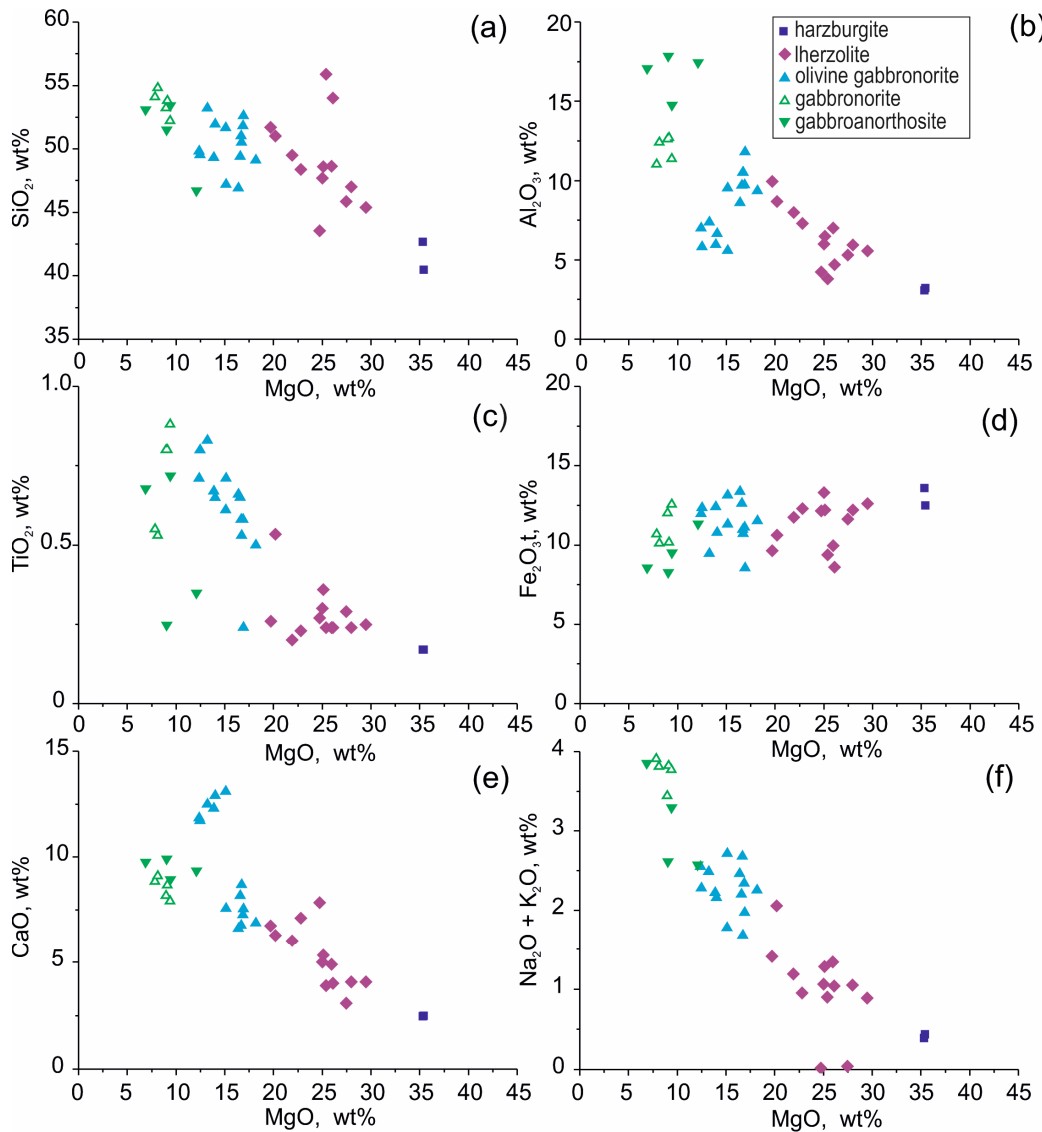

**Figure 12.** Petrochemical diagrams of MgO–SiO$_2$ (**a**), MgO–Al$_2$O$_3$ (**b**), MgO–TiO$_2$ (**c**), MgO–Fe$_2$O$_3$t (**d**), MgO–CaO (**e**), and MgO–(Na$_2$O + K$_2$O) (**f**) systems of Kovdozero, Gorely, and Yudom–Navolok islands massifs.

Mt. Generalskaya, Kivakka, and Kovdozero intrusion rocks exhibit the same inclined type of chondrite-normalized REE spectra (Figure 16a,c,e) with a slight excess of LREE over HREE (La$_n$/Lu$_n$ = 3–10); however, the intrusion rock types have a different degree of differentiation: the smallest degree is for Mt. Generalskaya rocks, a medium degree is for the Kovdozero massif, and the largest degree for the Kivakka intrusion. The main factor responsible for the differentiation of the spectra is olivine control; the inclined pattern of the spectra with enrichment in LREE over HREE is controlled by the excess of plagioclase and augite phases over orthopyroxene. Spider diagrams (Figure 16b,d,f) indicate enrichment in lithophile (Rb, Ba) and high-charge (Th, U) elements for all intrusions. One distinctive feature of the rocks analyzed is negative Nb–Ta and positive Sr anomalies; Kovdozero massif rocks also display a negative Sr anomaly. Diagrams of normalized REE spectra for the Burakovsky Pluton are constructed separately for different zones and two blocks (Figure 17). The rocks of dunite, peridotite, gabbronorite, and pigeonite-bearing gabbronorite zones, as well as Marginal Series rocks, exhibit the same type of inclined REE spectra with a similar excess of LREE over HREE (La$_n$/Lu$_n$ = 2–13) and a poorly defined positive Eu anomaly in gabbronorites. Rocks from the pyroxenite zones of the

Burakov–Shalozero and Aganozero Blocks are inclined less markedly than other rocks and differ considerably from each other: in the former block, they are enriched in REE and their spectra are inclined more substantially than those of the latter block (Figure 17b). The differences are due to a difference in phase composition: the Aganozero Block is dominated by orthopyroxene, while the Burakov–Shalozero Block by clinopyroxene. Dyke rocks are most similar to pigeonite-bearing gabbronorite zone rocks but contain more REE (Figure 17d).

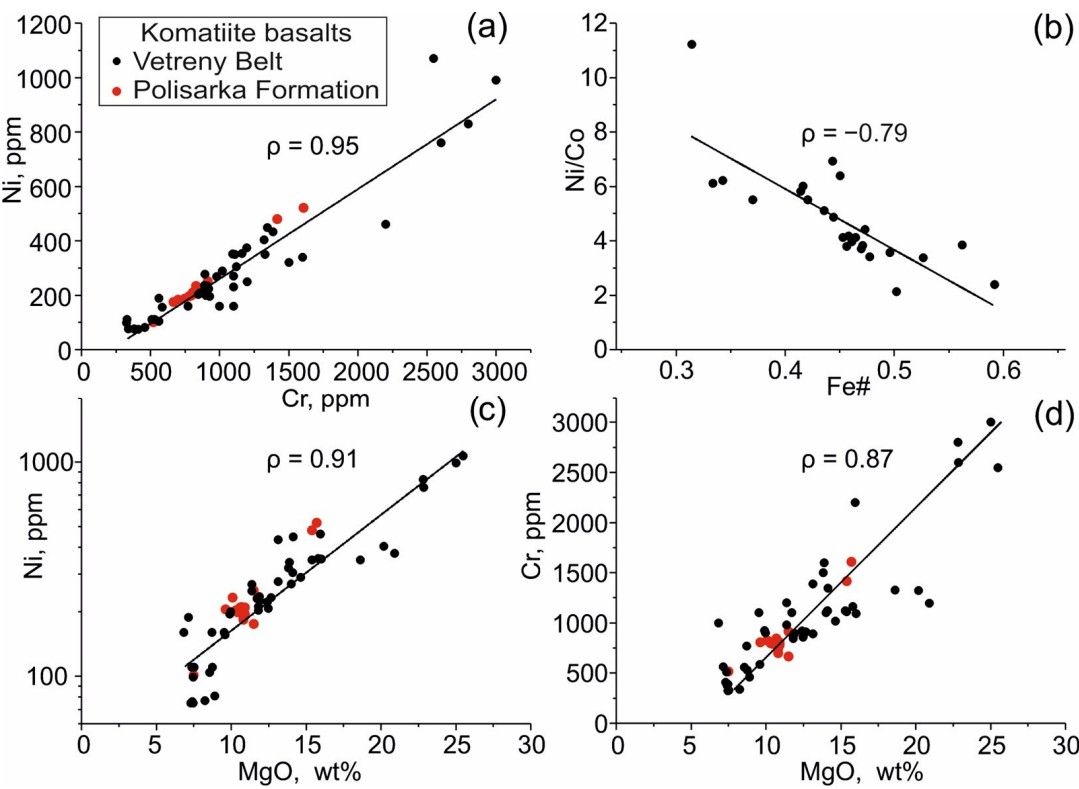

**Figure 13.** Correlation diagram showing Ni–Cr (**a**), Ni/Co–Fe# (**b**), Ni–MgO (**c**), Cr–MgO (**d**) of komatiitic basalts from Imandra–Varzuga and Vetreny Belts. Fe# = $Fe_2O_3t/(Fe_2O_3t + MgO)$. ($\rho$)—Pearson's correlation coefficient.

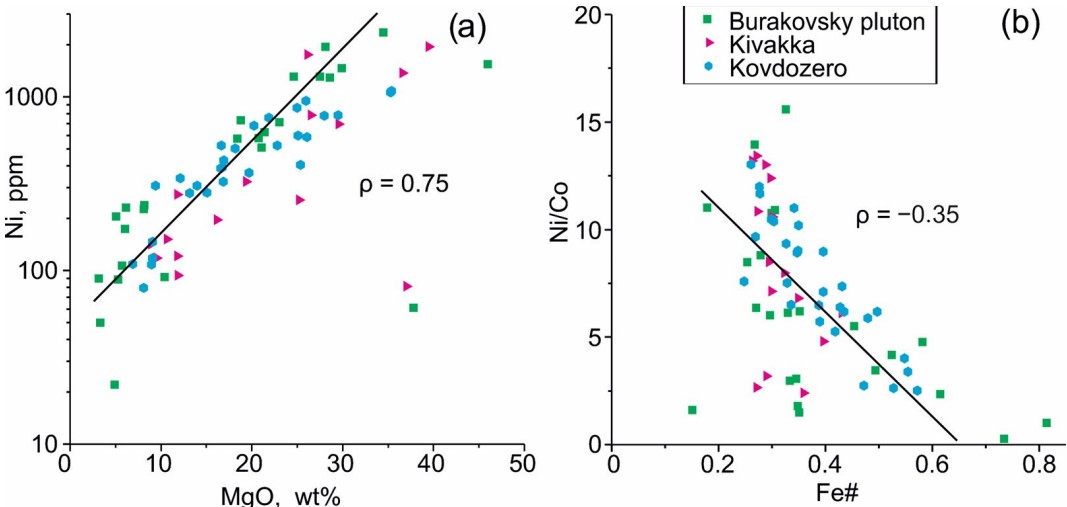

**Figure 14.** Correlation diagrams showing Ni–MgO (**a**), and Ni/Co–Fe# (**b**) of Burakovsky Pluton, Kivakka intrusion, and Kovdozero massif. ($\rho$)—Pearson's correlation coefficient.

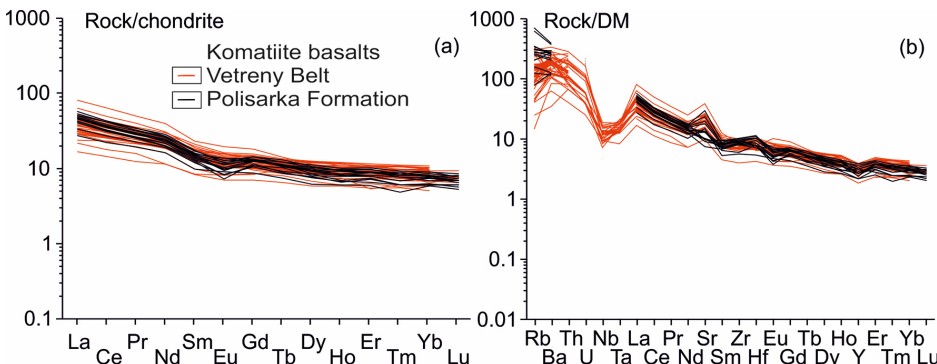

**Figure 15.** Spider diagrams for komatiitic basalts: (**a**) plots of REE distribution normalized to C1 chondrite [48]; and (**b**) plots of incoherent element distribution normalized to depleted mantle (DM) composition [49].

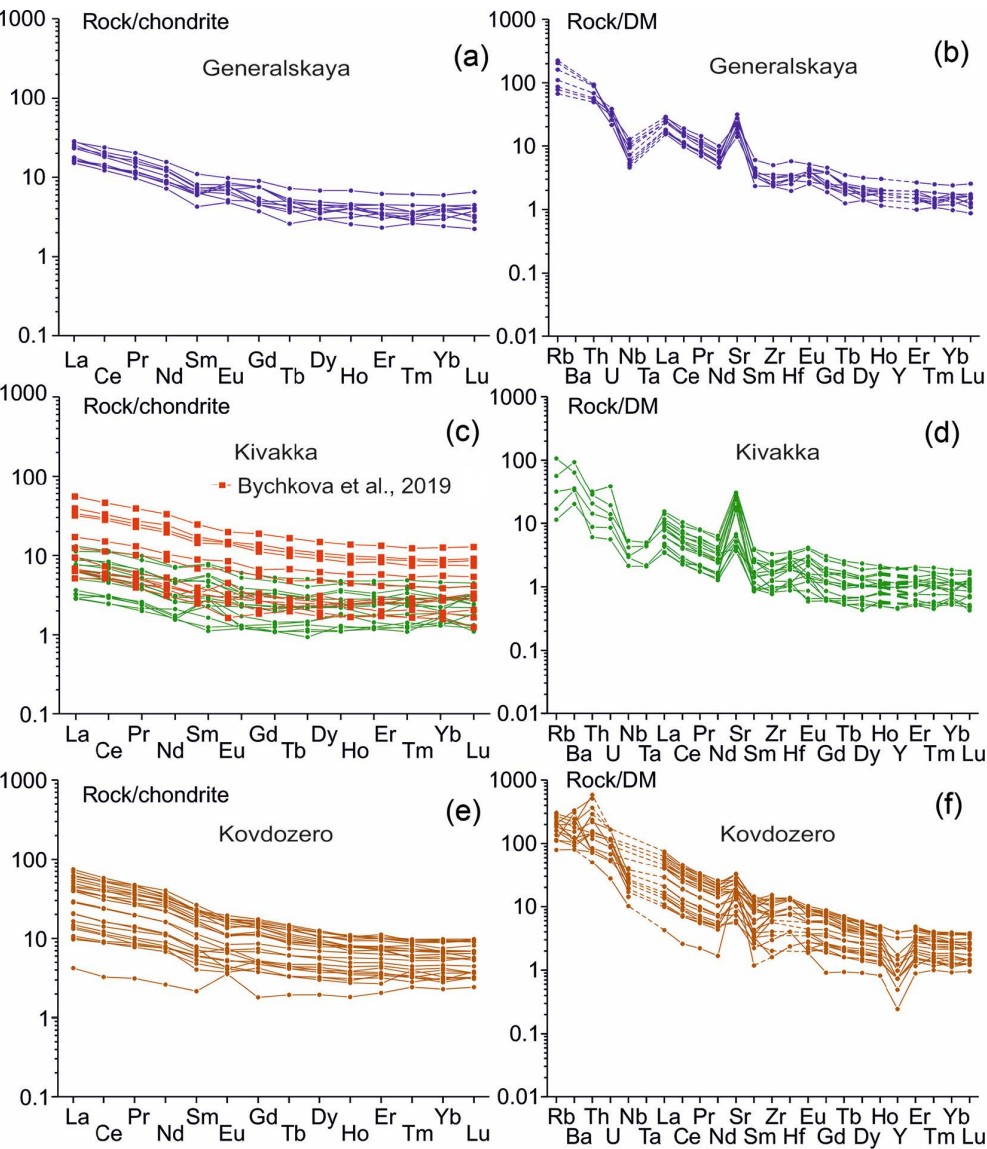

**Figure 16.** Spider diagrams for rocks of Mt. Generalskaya (**a**,**b**), Kivakka (**c**,**d**) [27], and Kovdozero (**e**,**f**) intrusions. The REE distribution normalized to C1 chondrite [48], and the incoherent element distribution normalized to depleted mantle composition (DM) [49].

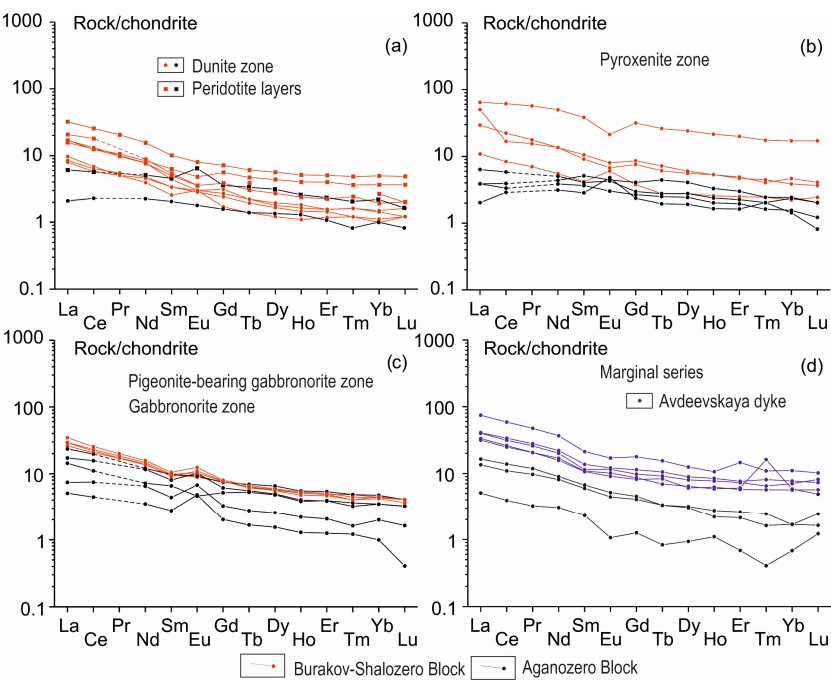

**Figure 17.** Chondrite-normalized [48] REE distribution of Burakovsky Pluton: dunite zone and peridotite layers (**a**); pyroxenite zone (**b**); pigeonite-bearing gabbronorite and gabbronorite zones (**c**); marginal series rocks (**d**). The REE distribution of Avdeevskaya dyke rocks is also shown (**d**).

The $La_n$–$Sm_n$ plot (Figure 18) shows a direct correlation for intrusive and volcanogenic rocks. Pearson's correlation coefficient ($\rho$) varies from 0.95 to 0.99 for Kovdozero and Kivakka intrusion rock and from 0.77 to 0.78 for Burakovsky Pluton rock. An intermediate $\rho$ value is shown by Monchepluton rocks and Vetreny Belt komatiitic basalts (0.84 and 0.86, respectively). The smallest scatter of composition points was obtained for Kivakka, while the greatest scatter was obtained for two Burakovsky Pluton blocks. Lower $\rho$ values are partly due to metamorphism.

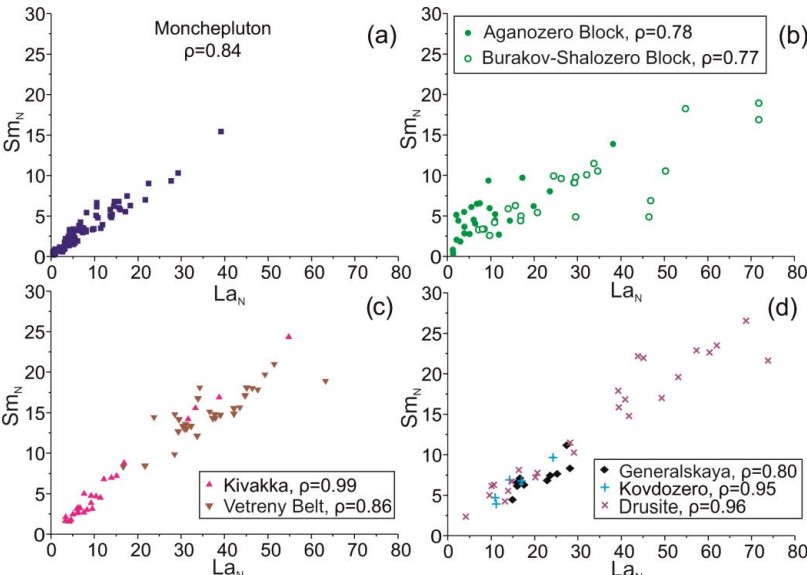

**Figure 18.** Correlation diagram $La_n$–$Sm_n$ of Monchepluton (**a**); Burakovsky Pluton (**b**); Kivakka intrusion and Vetreny Belt komatiitic basalts (**c**); Mt. Generalskaya intrusion, Kovdozero and other massifs of "Drusite Complex" (**d**). ($\rho$)—Pearson's correlation coefficient.

## 5. Discussion

### 5.1. Comparative Analysis of Geochemical and Isotopic Data

Our results for layered intrusions in the Kola and Karelian regions, reported in Sections 2 and 4, indicate that the intrusions differ not only in age but also in morphology, internal structure, geochemistry, and differentiation trends.

The cumulative MgO–SiO$_2$ diagram (Figure 19a) shows considerable overlapping of layered intrusion rocks in the MgO region of <25 wt% and differences in the MgO region of >25 wt%. The ultramafic rocks of Monchepluton contain higher MgO concentrations than medium concentrations in the Kivakka intrusion and the smallest concentrations in the Burakovsky Pluton. This difference is due not only to differences in the olivine, ortho- and clinopyroxene phase ratio in the rocks, but also differences in the Mg content of olivine. The rock field of Mt. Generalskaya is overlapped by the rock fields of megacycles IV and V of Monchepluton (Figure 19a,b), as well as the rock fields of the Kovdozero and other massifs of the "Drusite Complex" by the field of komatiitic basalts, supporting their sub-volcanic formation pattern.

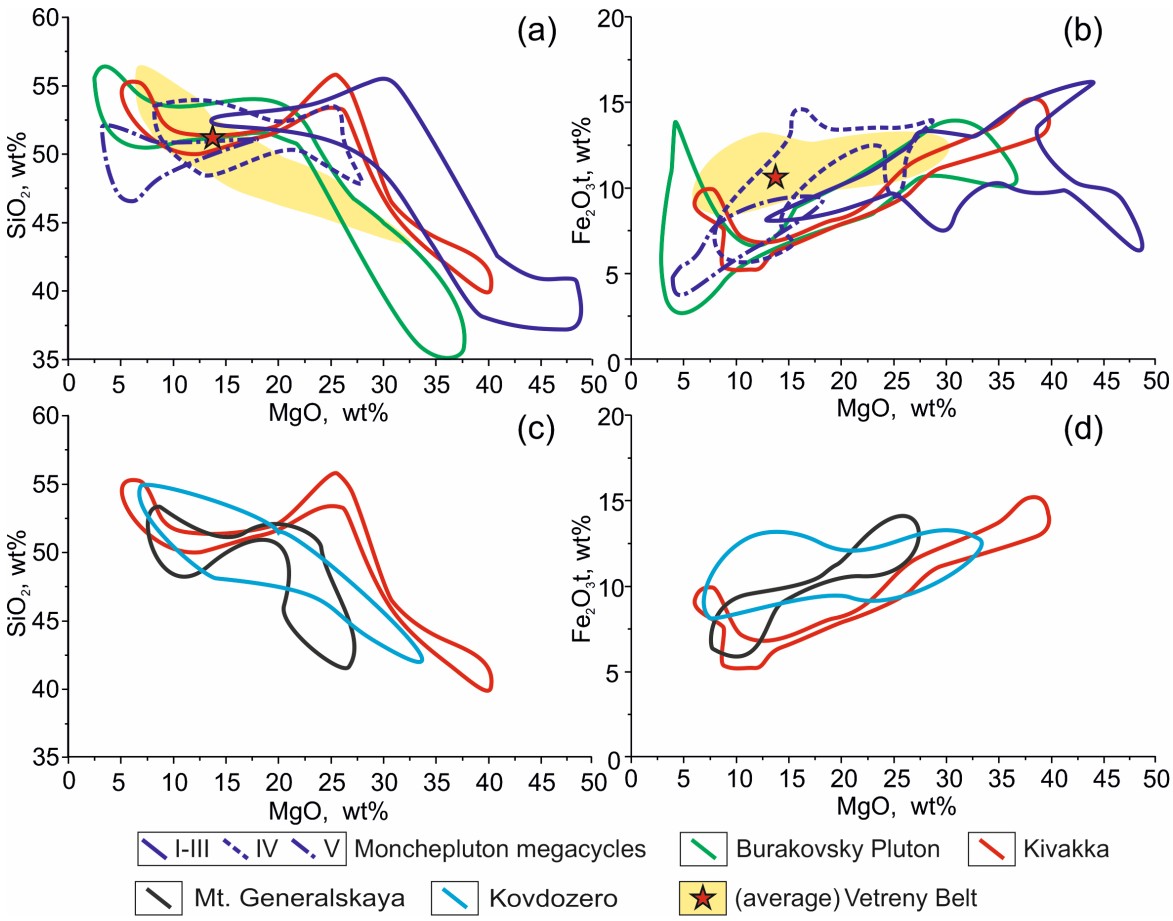

**Figure 19.** Rock composition fields of Monchepluton [1] and Burakovsky Pluton, as well as Kivakka, Mt. Generalskaya and Kovdozero intrusions on the petrochemical diagrams of SiO$_2$–MgO (**a,c**) and Fe$_2$O$_3$t–MgO (**b,d**). The complementary field and average composition of Vetreny Belt komatiitic basalt are also shown.

The MgO–Fe$_2$O$_3$t diagram (Figure 19b,d) shows the complex configuration of fields for the Monche- and Burakovsky-Plutons against a general decrease in components during differentiation. Accepting the one-chamber Kivakka intrusion as a model of a closed system, we assume periodical variations in magma crystallization conditions (oxygen fugacity, etc.) provoked by the disturbance of the closed structure of the system for the Monche- and

Burakovsky-Plutons. One reason for that is the supply of an extra portion of magma into a magma chamber from deep sources.

To better understand the above differences, let us reassess the results of earlier isotopic studies of the Kivakka intrusion and the Burakovsky Pluton [17].

The diagram (Figure 20) shows variations in Nd concentrations and the primary $\varepsilon_{Nd}$ (T) ratio for the generalized vertical sections of the above intrusions. For the Kivakka intrusion, the primary $\varepsilon_{Nd}$ ratio (T) varies from $-0.5$ to $-2.0$. The asymmetrical distribution of Nd concentration with an increase in the lower (2.5 ppm) upper (9.5 ppm) contacts was also revealed. Nd concentration is controlled by ortho- and clino-pyroxenes, which are more abundant in Upper Marginal Series rocks. This evidence does not contradict with the model of Ya. Bychkova et al. [27], which shows that magma crystallization and differentiation in the Kivakka chamber were characteristic of a closed system. The chamber was separated from host rocks by Marginal Series rocks at the early stage of magma crystallization. The Aganozero Block of the Burakovsky Pluton displays a different pattern. The Lower and Upper portions of the sequence clearly vary in primary $\varepsilon_{Nd}$ (T) ratios from 1 to $-1.3$ and from $-1.6$ to $-2.0$, respectively. The interface (340 m) coincides with the contact between the lower and upper sub-zones of the gabbronorite zone [19]. A high (2.7–2.8) primary $\varepsilon_{Nd}$ (T) ratio value in fine-grained gabbronorite (sample C-68/130) was obtained at the same interface. Variation in Nd concentration reveals three independent inclined trends; each trend begins (from the base upwards) with low values and ends with high values. The boundary between the second and third (from the base) trends is also located at a depth of 340 m. These data indicate the disturbance of magma crystallization presumably provoked by the injection of another portion of magma with different isotopic characteristics.

Some time ago, we analyzed variations in the primary $\varepsilon_{Nd}$ (T) ratio of layered intrusion rocks (including those described above) in the massifs of gabbro-anorthosite complexes (The Main Ridge, Pyrshin), in the massifs of the "Drusite Complex", western White Sea region, and in Vetreny Belt gabbronorite dykes and komatiitic basalts [1]. The bulk of analytical data, including those on volcanics, range from $-1$ to $-3$ $\varepsilon_{Nd}$ (T). Maximum scatter was obtained for layered intrusions and gabbro-anorthosite massifs, while minimum scatter was obtained for dykes, volcanics, and "Drusite Complex" massifs. Host plagiogneisses in Archean complexes display the lowest primary $\varepsilon_{Nd}$ (T) ratio values varying from $-3.5$ to $-18.1$. Both anomalous and mantle primary $\varepsilon_{Nd}$ (T) ratio values were obtained for Monchepluton dunites and chromitites, both for its feeding magmatic channel and for Pados-Tundra dunites and chromitites (1 to 3).

The $\varepsilon_{Nd}$ (T)–$^{87}Sr/^{86}Sr$ plot for combined analysis of two isotopic Sm–Nd and Rb–Sr systems in the rocks of layered intrusions is used (Figure 21). The mafic rocks of the Kivakka intrusion, Monchepluton (megacycle IV), and Monchetundra massif (Upper and Lower zones), as well as a few analyses of Burakovsky Pluton websterite, occupy a combined separate field (I) with negative $\varepsilon_{Nd}$ (T) values of $-0.3$ to $-2.5$ and moderate $^{87}Sr/^{86}Sr$ ratios of 0.7017 to 0.7027.

The rocks of both Burakovsky Pluton blocks typically display a highly non-uniform and contrasting distribution of analytical points. A small group of rocks (II) shows positive $\varepsilon_{Nd}$ (T) values of 0.4–0.9 in serpentinite and 2.7–2.8 in fine-grained gabbronorite (sample borehole 68/130), as well as elevated $^{87}Sr/^{86}Sr$ values of 0.7033. The bulk of the mafic rocks (III, IV) exhibit gradually increasing negative $\varepsilon_{Nd}$ (T) values of $-0.3$ to $-2.5$ and $^{87}Sr/^{86}Sr$ values of 0.7028 to 0.7038. The highest $^{87}Sr/^{86}Sr$ values (up to 0.7039) were obtained for plagioclase from the gabbronorites of the Aganozero and Burakov–Shalozero Blocks.

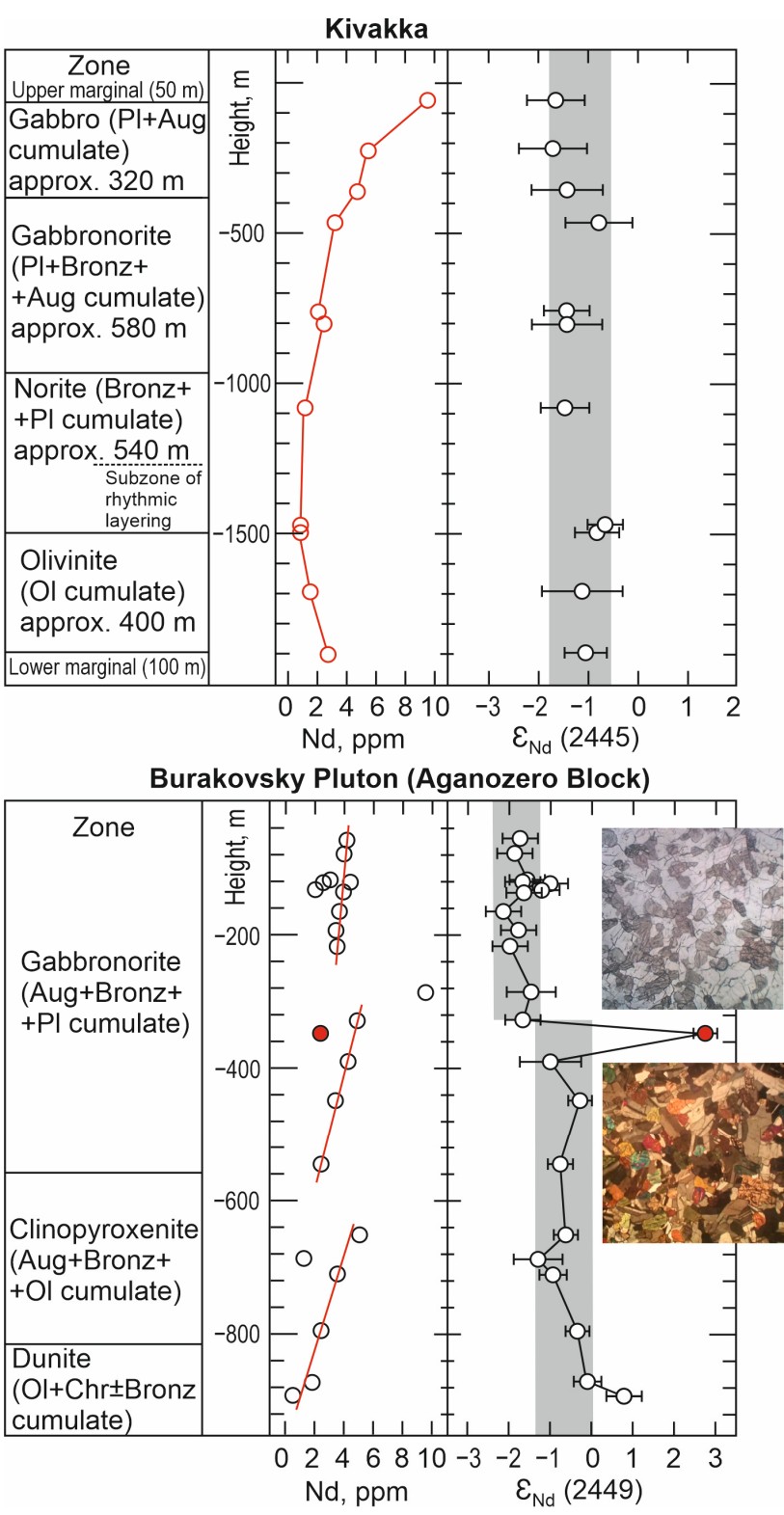

**Figure 20.** Variations in Nd content and initial ratios of $\varepsilon_{Nd}$ (T) in rocks according to vertical cross-section of Kivakka intrusion and Burakovsky Pluton (Aganozero Block) (modified after study) [17]. Photomicrographs of thin section of fine-grained gabbronorite, sample C-68/130.3 (polarized and natural light). Mineral abbreviations: Pl—plagioclase; Aug—augite; Bronz—bronzite; Ol—olivine; Chr—chromite.

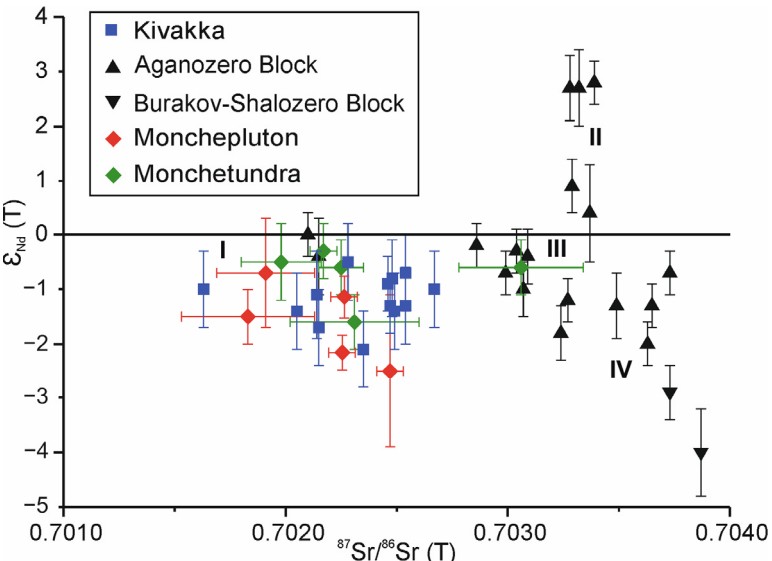

**Figure 21.** Variations in the initial ratio of $\varepsilon_{Nd}$ (T) and $^{87}Sr/^{86}Sr$(T) in rocks of Paleoproterozoic layered intrusions: Monchepluton, Burakovsky Pluton, Monchetundra, and Kivakka intrusions. Data from [8,17,19,50,51].

The above diversity could be due to various factors. Two major hypotheses were proposed for the rocks of group I: a model of the melting of an enriched lithospheric source and a model of a mantle plume interacting with Neoarchean Crust [17]. Model calculations, made by independent research teams [17,51], show that the second model is more preferable. In some cases, mantle labels for Monchepluton and Pados–Tundra dunites and chromitites [52] indicate that homogenization was incomplete. Layered intrusions are assumed to display a higher degree of interaction than volcanic rocks. We also assume that contamination and assimilation with Archean amphibolite–gneiss host complexes took place locally during magma uplift. A crustal component could actually exist, as evidenced by frequently occurring relics of ~2.7 Ga zircon and the results of isotopic analysis of S, which will be discussed below.

The unique anomalous isotopic characteristics of the rocks and minerals of the Burakovsky Pluton are hard to explain based on crustal matter contamination alone. The presence of phlogopite in peridotites, quartz, biotite and K-feldspar associations in pigeonite-bearing gabbronorites, as well as inclusions of quartz-carbonate composition in pyroxenites from the zone of the same name, indicate post-magmatic metasomatism and the addition of Sr-enriched fluids. There are carbonate-bearing serpentinites in the Aganozero Block, at a depth of up to 70 m. Hydrotalcitic-lizarditic serpentinites with chlorite, carbonate, and high concentrations of acid-dissolvable forms of Ni, Mg, and Fe occur at deeper horizons [53]. The high $^{87}Sr/^{86}Sr$ ratio of 0.7033 upon dunite alteration suggests that enrichment in Sr was primarily due to carbonate. To solve this problem, further studies are needed.

The U–Th–Pd isotope age of zircon from gabbro-pegmatite cutting Avdeevskaya Dyke rocks (Table 1) has led us to conclude that metamorphism took place about 2.0 Ga.

### 5.2. S Isotope Analysis

The isotope composition of sulfur is an essential indicator of sulfide ore formation. More data on Monchegorsk Ore District intrusions were obtained by two independent research teams [54,55]. Ore and rock samples from boreholes and quarries of the following deposits and ore occurrences were analyzed: Nittis–Kumuzhya–Travyanaya (NKT), Sopcha, Terrasa (Nyud City) and Nyud-II (Monchepluton), "Nickel Creek" (Volchya-Tundra massif), and rocks in Monchetundra (borehole M-1/749 and 955 m). The isotopic composition of $\delta^{34}S$ varies from −0.30 to +1.14‰ for sulfide ores and from −0.64 to +1.94‰ for host rocks, coinciding with the range ($\delta^{34}S$ = 0 ± 2‰) for sulfur of mantle origin. However, negative

values were obtained for $\Delta^{33}$S: $-0.06$ to $-0.26$‰ for ore samples and 0.10 to $-0.23$‰ for host rocks. These results indicate traces of mass-independent sulfur fractionation, and are not consistent with its mantle source ($\Delta^{33}$S $= 0.00 \pm 0.03$‰) [54,55]. The most probable mechanism, which could explain the presence of isotopically anomalous sulfur in Monchepluton ores and rocks is the contamination of crustal matter by magma in intermediate chambers. Contamination could have taken place at the early stages of intrusion formation, providing sufficient time for isotopic homogenization before the separation of sulfide melt from its silicate matrix. Sulfur could have been supplied from the metasedimentary rocks of Archean greenstone belts unexposed by erosion and primarily enriched in the sulfur of sulfate minerals.

*5.3. Layered Intrusion Formation Pattern*

At the 2.50 Ga boundary, the long-lived mantle plume was uplifted and mantle magma beneath Baltica Paleocontinent was generated in what is now the Kola–Lapland–Karelian Province of the oldest eastern portion of the Fennoscandian Shield [9]. As a result of the uplift of abundant primary mantle magma into the lower earth crust, it was heated and disrupted; paleorift-related systems, filled with various sedimentary and volcanogenic rock complexes, then formed.

The large-scale interaction of primary magma with the granulite–eclogite complex of the lower crust triggered the formation of deep-seated komatiitic basaltic magma chambers. Magma was then uplifted into intermediate and magma chambers at two stages—2.50 and 2.45 Ga—interacting locally with crustal matter and enriching it heterogeneously in sulfur. In addition to komatiitic basaltic magma chambers, basaltic magma chambers, with which numerous dolerite dyke swarms with isotopic mantle labels are associated, and coeval granitic massifs were derived.

Magma chambers formed at various upper crustal levels. They were filled in a non-uniform manner, triggering the formation of intrusive massifs varying in internal structure and ore mineralization pattern. One- and multi-chamber intrusions were identified. One-chamber intrusions seem to have been filled within a short period of time; the equiponderous crystallization in them was based on a closed-system scheme. Kivakka is a demonstrative example of such an intrusion. Monche- and Burakovsky plutons are multi-chamber intrusions. They were filled repeatedly in a pulse-like manner, meaning that the rock sequence was often disturbed due to variable crystallization conditions. Each chamber or sub-chamber could have its own rock combination, mineral phase composition, and mineralization. One feature of such layered intrusions is the washing-out of a cumulative layer by a new portion of hotter magma and the disturbance of crystallization equilibrium.

Layered intrusions consist of rocks produced by the crystallization and differentiation of high-Mg magma. Geochemical and isotopic data show that the composition of parent magma for layered intrusions is similar to that of komatiitic basalts. It should be noted, however, that the eruption of komatiitic basalts took place later (2.41 Ga) than the emplacement of layered intrusions (2.50–2.45 Ga). It seems that parent magma for layered intrusions was richer in Mg and contained ore matter. The composition of mantle reservoirs is also indicated by the deep xenoliths of 2.47–2.41 Ga spinel peridotites and pyroxenites in Paleozoic explosion pipes and explosive dykes in the White Sea region [56]. The composition of the xenoliths and the primary $\varepsilon_{\text{Nd}}$ (T) ratio ($-0.8$ to $-2.5$) are comparable with those of similar rocks in layered intrusions.

Most layered intrusions consist of mildly metamorphosed rocks. However, some have been partly subjected to multiple, genetically diverse metamorphic processes. The ultramafic rocks of the Burakovsky Pluton have been intensely metasomatized locally by a fluid flow presumably coming from a deep source of granitic origin. "Drusite complex" massifs have experienced high-temperature granulite-facies metamorphism, which provoked the formation of corona textures during the Svecofennian Orogeny (1.91 Ga).

Mt. Generalskaya is a fissured intrusion partly exposed by glacial erosion during the global Huronian Glaciation. The large southwestern portion of the intrusion, overlain by

a thick cover of Pechenga Complex conglomerate and volcanogenic rocks, is still poorly understood. We believe that the unexposed portion of the intrusion contains no olivine-enriched rocks (e.g., dunites and peridotites) and near-bottom ores enriched in nickel and copper sulfides.

Kivakka is a classical type of layered intrusion with a complete combination of rocks from dunites to leucogabbro. It is a one-chamber intrusion filled with magma as a result of a one-act process. Marginal Series rocks formed a barrier between magma and host rocks, while exocontact hornfels zones did not occur. Magma in this chamber crystallized, as did magma in a closed system. As temperature decreased, crystallization was followed by the consecutive crystallization and accumulation of olivine, ortho- and clino-pyroxene, and plagioclase. Granophyre, a quartz-bearing rock which occupied a small portion of the upper zone of the chamber, was a final product. The crystallization and differentiation of the Kivakka intrusion can be easily reconstructed based on model calculations using COMAGMAT program [27].

The Burakovsky Pluton displays a more complex structure. We propose a model of its formation based on reassessment of available geological evidence and geochemical and isotopic data. The Burakov–Shalozero and Aganozero Blocks should be interpreted as two sub-chambers rather than tectonic blocks. According to the model, primary high-Mg magma was uplifted at an early stage and both sub-chambers were partly filled. Because olivine cumulates, occurring as dunites and peridotites, were abundant, the sub-chambers were filled repeatedly at short intervals. Subsequent portions of magma were supplied from an intermediate chamber originally into the Burakov–Shalozero and the Aganozero sub-chambers, forming variably thick chromite horizons and compositionally diverse pyroxenite and gabbronorite zones. At the final stage, magma was supplied mainly into the Burakov–Shalozero sub-chamber. The deposition of magnetite-bearing rocks indicates considerably increased oxygen fugacity and crystallization under closed-system conditions. At the end of this stage, magma began to fill a sub-vertical fault system, forming the Avdeyevskaya Dyke. Today, the time taken for the formation of the Burakovsky Pluton cannot be calculated due to the scarcity of isotopic data.

The Monchepluton intrusion is unique with its complete combination of sulfide Cu-Ni, Cr and low-sulfide PGE ore deposits; it is a layered two-chamber intrusion with a variable combination of rocks and deposits [7,8,10]. It was filled with magma periodically for about 8 million years as a result of several intrusion cycles. Portions of magma intruded before the complete solidification of a previous portion; as a result, cumulates were washed out or brecciated, though no phase relationship formed. The composition and ore potential of magma were controlled by their preliminary differentiation in deep-seated and medium-depth chambers, giving rise to various forms of ore mineralization. Monchepluton is characterized by the presence of silicate and ore-silicate pegmatoid bodies occurring at the various levels of the rock sequence. They indicate originally high fluid concentrations in magma and periodical fluid accumulation. At the final stage, residual melts were squeezed out into contraction joints, forming mafic dykes. The age of the gabbropegmatites and dykes is similar to the average age of intrusive host rocks (2502 ± 5 Ma).

The big Main Ridge Gabbro-anorthosite Complex, similar in isotopic age, was forming for a long time by the consecutive intrusion of Monche-, Chuna–, Volchya–, and Losevo–Medvezhya Tundra massifs in the deep fault area [1,8]. The massifs consist mainly of mafic rocks (gabbronorites, norites, and anorthosites) injected by ultramafic rock (dunite and harzburgite) bodies. The formation of the biggest massif, known as Monchetundra, falls into two stages: 2507–2496 and 2476–2471 Ma. Gabbro-pegmatites formed later: 2456–2445 Ma; the timing of their formation coincides with the intrusion of Imandra–Umbarechka Complex massifs (2456–2437 Ma).

The Kovdozero and other "Drusite Complex" massifs occur as shallow-depth dominantly lenticular and horseshoe-shaped sub-volcanic bodies. Their intrusion was affected by the active movement of Archean host rock complexes. One of their distinctive features is the presence of corona structures of magmatic and metamorphic types [39]. Their general

rock combination is similar to that in layered intrusion. Some differences due to shallow-depth crystallization conditions occur: a lower degree of differentiation, the absence of dunites and chromitites, and the olivine-bearing and two pyroxenes peridotites that seldom occur in layered intrusions.

## 6. Conclusions

The Paleoproterozoic layered intrusions of the peridotite–pyroxenite–gabbronorite complex, occurring in the eastern Fennoscandian Shield, were mainly studied using geochemical methods and data on isotopic U–Pb, Sm–Nd, and Rb–Sr systems. The intrusions with various structures, rock varieties, and degrees of differentiation, such as Mt. Generalskaya, Kivakka, Kovdozero, and the Burakovsky Pluton, as well as Ti-depleted komatiite–basalt volcanics, were chosen for the study.

(1) In terms of internal structure, there are one- (Kivakka) and multi-chamber (the Burakovsky Pluton and previously studied Monchepluton) intrusions;

(2) In a one-chamber intrusion (Kivakka), magma crystallized and differentiated in the same way as that in a closed system with insignificant accumulation of quartz-bearing rocks (or granophyres) and a Ti-rich ore phase represented by Ti-magnetite;

(3) The Burakovsky Pluton is consistent with a model in which magma repeatedly filled two contiguous sub-chambers, depositing various amounts of compositionally diverse dunites, peridotites, gabbronorites, and rock-forming minerals. At the final stage of magma chamber closure, another portion of magma compressed the sub-vertical faults, forming a large dyke system;

(4) The shape and internal structure of "Drusite Complex" massifs in the western White Sea region were mainly affected by the active migration of the enclosing frame, which prevented the accumulation of sulfide-rich ore horizons;

(5) The formation of layered intrusions was greatly contributed to by two factors: tectonics and the presence of intermediate magma chambers. The tectonic migration of country rock blocks resulted in the redistribution of channels for magma transfer from intermediate magma chambers. Preliminary magma differentiation in magma chambers was responsible for a variety of rocks in some magma chambers;

(6) During the uplift of primary mantle magmas, the contamination and assimilation of rock units in the Neoarchean crust evolved locally in deep-seated and intermediate magma chambers, while country rocks in the endocontact zones of magma chambers are mildly contaminated.

**Supplementary Materials:** The following supporting information can be downloaded at: https://www.mdpi.com/article/10.3390/min13050597/s1, Table S1: Major, Trace and Rare earth elements composition of the Mt. Generalskaya intrusion; Table S2: Major, Trace and Rare earth elements composition of the Kivakka intrusion; Table S3: Major, Trace and Rare earth elements composition of the "Drusite Complex" rocks (Kovdozero massif)

**Author Contributions:** The authors wrote the paper together. V.F.S.: field works, studies, discussions, and conclusions; A.V.M.: field work, maps and diagrams, studies, and interpretations; A.V.C.: field work, studies, and discussions. All authors have read and agreed to the published version of the manuscript.

**Funding:** The Geological Institute of the Kola Science Center of Russian Academy of Sciences (GI KSC RAS 0226-2019-0053).

**Data Availability Statement:** The data presented in this study are openly available online at Supplementary Materials.

**Acknowledgments:** We are grateful to A.A. Kremenetsky (Institute of Mineralogy, Geochemistry and Crystal Chemistry of Rare Elements, Moscow, Russia) for the financial support for the studies. The authors express gratitude to anonymous reviewers and members of the Editorial staff who helped us improve the presentation of our results.

**Conflicts of Interest:** The authors declare no conflict of interest.

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
