# Peer review of "Layered Intrusions of Paleoproterozoic Age in the Kola and Karelian Regions"

_minerals, doi:10.3390/min13050597_

Round 1

Reviewer 1 Report

The manuscript is devoted to internal structure and petrogeochemical and isotopic characteristics of a Paleoproterozoic mafic-ultramafic layered intrusions in Kola and Karelia (SW Russia). Obtained results based on a compilation and analyses of previous published data and author’s new compositional data. Concluding the results, the authors proposed some petrological models for the forming of Paleoproterozoic layered intrusions.

The article devoted to these problems may be published in Minerals.

However, I consider it necessary to give a few comments that will probably help improve the manuscript.

Title

Perhaps the title of the manuscript should contain information about the results obtained - composition, age, petrological models of formation?

Line 64-65. “Kovdozero” repeated twice.

Line 68-69. The authors have stated their aim as “construct a petrologic model showing the formation of Proterozoic layered intrusions”. But in Discussion and Conclusions authors give several separate models for each intrusion, and did not make a general unified model for all intrusions.

Figure 1. It would be better if the authors indicate (for example, circle) the studied intrusions. It's also better to move the legend to the bottom left corner

Figure 3. According to bedding given on a map, the layers of Kivakka intrusions falls to NW. So the lower part of intrusion should be at its SE margin and the upper part of intrusion should be at its NW margin. It seems that the positions of the lower marginal zone and the upper marginal zone are mistake in the figure.

Lines 420-430, Figure 7. Authors discuss the composition of Polisar formation but the legend of Figure 7 do not contain the Polisar fm.

Section 5.3.  Authors discuss the models of Monchepluton formation but they did not give early any information about its geology, age and composition. And against the authors do not give any model for Mt. Generalskaya intrusion which are detail describe in earlier sections of text.

Conclusions. Some information in this section repeat an information from Introduction section. And many information from this section may be moved to the Discussion section. It will be better if the only brief conclusion sentences will be in this section.

Reviewer 2 Report

Title is Ok

Line 14, abstract: Some of your intrusions, e.g. Mt. Generalskaya, have anorthosite such as the case shown in Fig. 2. Therefore, you need to report this rock variety in the abstract.

Abstract is not bad but it needs some modifications as I shown in the attached annotated file (pdf).

Line 24: Your keywords need to include: magma generation or evolution.

Line 82, Fig. 1: Coordinates (latitudes and longitudes) for the map are missing. In the legend, you need to emphasize about the oldest and the youngest rock units (Archean complex and Celadonides, respectively). Please specify and give a simplified lithology and possible age.

Line 84, Fig. 2: What does the marginal zone made of?. Is it a peripheral zone with a certain rock composition or a chilled margin?!.

Line 96, footnotes of Table 1: You need to write the name of the analysing techniques in full.

Line 97: When you describe zircon related to metamorphism, you should say metamorphic zircon and not metamorphosed zircon. Metamorphosed is used as an adjective for a rock, not for a mineral.

Line 127: do not use mineral abbreviation for the first time unless you write the name in full, e.g (forsterite, Fe) and ferrosilite (fs). You need to do the same for anorthite (An) in line 170.

Line 134: you mentioned about streaky-disseminated type. You need to write the name of the analyzing techniques in full.

Line 149, Fig. 3: Again, latitudes and longitudes are missing.

Line 177: Probably you mean sericite and saussurite. If the rock is albitized then the calcic plagioclase can be replaced by albite.

Line 195, Fig. 4: Latitudes and longitudes are still missing. I believe that the sequence of these ultramafic and mafic rock varieties is upside down, isn't it?. If so, you need to reverse it.

Line 216: What is the volume and size of these plagioclase websterite in the peridotite?. Is it abundant, randomly scattered or show even distribution?.

Line 323: Chromite in the form of crystals or grains (mostly as Cr-spinel) do not alternate with the host rocks and most probably you mean chromitite ore even though it occurs as thin or micro-layers.

Line 242: You need to insert a reference or two in order to support the metsaomatic alterations of the cumulate rocks of an earlier igneous origin.

Line 255: Are you sure you have two varieties of magnetite, one normal and one Ti-bearing?. Are there any previous work that contains EMPA for them to confirm?.

Line 314: I wonder if it is a "pegmatite" of felsic composition and probably you mean pegmatitic gabbro for the very coarse-grained mafic differentiate.

Line 321, Fig. 6: It needs geographic co-ordinates same as the rest of the geological maps you use in the manuscript. In the legend, place of these ultramafic rocks is puzzling?. They are supposed to be country rocks of the ultramafic and mafic intrusives of the massif. So, they must not stand intermittent to the igneous sequence.

Line 343: Plagioclase is a felsic mineral and uncommon to be brown in colour. Are you sure that coloration is not attributed to overgrowth or fine mineral inclusions?. Can the post-magmatic alteration process result in this?.

Line 396: Always write REEs as rare-earth elements.

Line 440, Fig. 7: If possible, use colours instead of open and closed circles as symbols. Almost all oxides show good correlation with oxides. Is it possible to show the regression value as R2?.

Line 459: Here and elsewhere, references are needed to support the observation or conclusion.

Line 493, Fig. 11: Is it total iron as ferrous (Fe2+)?. Also, don't place the legend between the figure and its caption. Move it to the top of figure.

Line 513, Fig. 12: Probably you mean anorthositic gabbro, please elaborate?.

Line Fig. 515, Fig. 13: In this figure, as well as some other figures (e.g. Fig. 14), you drafted a trend line for the correlation of MgO with transitional elements, which is good but it is recommended to insert the regression value as well.

Line 531, Fig. 15: They are both spider diagrams and only the normalizing reference is the difference, either chondrite or average mantle.

Line 533: Mount... Do not abbreviate when you start a sentence or a paragraph.

Line 625, Fig. 20: You need to report abbreviations of minerals you use in the figure in the figure captions, e.g. Ol, bronz, aug,....etc.

Line 669: Do you have evidence that the alteration of your ultramafic rocks represent post-magmatic metasomatism?. You need to emphasize and elaborate this. You mentiond about regional metamorphism in line 676. In this case, you need to distinguish between prograde metamorphic rock which is the serpentinite and its retrograde counterpart, which is rich in chlorite and Fe-rich carbonate  such as ankerite, if any.

Line 675: The issue of possible pegmatitic gabbro appears again in the discussion section and it needs careful handling of the nomenclature.

Line 765, conclusions: You still needs to support your conclusion that the course of igneous crystallization of the investigated intrusions practiced some contamination and assimilation. You need to elaborate this and show how they match with fractional crystallization as the principle process in the magma chambers that led to the intrusions.

Lines 783-785, conclusions: I do not agree about this conclusion. If you refer to the geological maps you use for the studied intrusions you can obviously notice that tectonics post-dates the emplacement and full solidification of the ultramafic and mafic rocks. It is hard to occur during the late stages of crystallization, and if so you need some evidence to support your hypothesis.

Line 824, reference list: Please re-edit the reference list carefully and follow the journal instruction to prepare it. Pay a special attention to the highlighted parts in which you sometimes abbreviate the journal names and sometimes not.

In the supplementary files, you need to subscript oxides, indicate the ppm as the concentration unit before the first element not beside it, and indicate if the (-) means not determined or not detected. Your data for the Mt. Generalskaya intrusion lack some trace elements and rare-earth of some. Any explanation for this (Table 1S)?.

Round 2

Reviewer 1 Report

The authors have done significant work to improve the manuscript

Author Response

Dear reviewer,

we thank You for your constructive comments comments, which allowed us to significantly improve the paper.

The paper has been carefully proof-read (especially newly added and rewritten parts of the Abstract, Discussion and Conclusions) by an English expert with a practice and sound experience in publishing in Minerals. The legends of the maps have been corrected, and the color has been added to the graphics.

Sincerely yours,

Valery F. Smol’kin and Artem V. Mokrushin

Reviewer 3 Report

The revisions are acceptable for this contribution. 

Author Response

(The authors gave the same response as above.)
